# N6-methyladenosine modification governs liver glycogenesis by stabilizing the glycogen synthase 2 mRNA

Xiang Zhang [1,2,8], Huilong Yin [1,3,4,8], Xiaofang Zhang[1], Xunliang Jiang[1], Yongkang Liu[1], Haolin Zhang[1], Yingran Peng[1], Da Li[1], Yanping Yu[1,5], Jinbao Zhang[1], Shuli Cheng[1,6], Angang Yang [3,4,7] & Rui Zhang [1,7] ✉

Hepatic glycogen is the main source of blood glucose and controls the intervals between meals in mammals. Hepatic glycogen storage in mammalian pups is insufficient compared to their adult counterparts; however, the detailed molecular mechanism is poorly understood. Here, we show that, similar to glycogen storage pattern, N6-methyladenosine (m6A) modification in mRNAs gradually increases during the growth of mice in liver. Strikingly, in the hepatocyte-specific Mettl3 knockout mice, loss of m6A modification disrupts liver glycogen storage. On the mechanism, mRNA of Gys2, the liver-specific glycogen synthase, is a substrate of METTL3 and plays a critical role in m6A-mediated glycogenesis. Furthermore, IGF2BP2, a "reader" protein of m6A, stabilizes the mRNA of Gys2. More importantly, reconstitution of GYS2 almost rescues liver glycogenesis in Mettl3-cKO mice. Collectively, a METTL3-IGF2BP2-GYS2 axis, in which METTL3 and IGF2BP2 regulate glycogenesis as "writer" and "reader" proteins respectively, is essential on maintenance of liver glycogenesis in mammals.

Hepatic glycogen maintains the blood glucose concentration between meals. In humans, defects in glycogenesis cause different types of glycogen storage diseases, which may lead to failure to thrive in some pediatric patients[1,2]. Intriguingly, under physiological conditions, the hepatic glycogen content increases gradually from birth to adulthood in numerous mammals, such as rats, rabbits, sheep, and rhesus macaques[3–5]. And the blood glucose concentration has a similar pattern[6]. However, the detailed mechanism and biological meaning of this phenomenon remain unclear.

Gys2, located at 12p12.1 in human, is conserved in chimpanzee, rhesus monkey, dog, cow, mouse, rat, chicken, and zebrafish. The protein encoded by this gene is liver glycogen synthase (GS), a key enzyme in glycogenesis, and catalyzes the addition of α−1,4-linked glucose to the growing glycogen chain. Mutations in this gene cause glycogen storage disease type 0 (GSD-0) in early childhood, with hypoglycemia and liver glycogen defect as symptoms[1,7]. However, little is known about regulation of *Gys2* expression.

[1]The State Key Laboratory of Cancer Biology, Department of Biochemistry and Molecular Biology, Fourth Military Medical University, Xi'an, Shaanxi 710032, China. [2]The Ministry of Education Key Lab of Hazard Assessment and Control in Special Operational Environment, Fourth Military Medical University, Xi'an, Shaanxi 710032, China. [3]The Henan Key Laboratory of immunology and Targeted Therapy, School of Laboratory Medicine, Xinxiang Medical University, Xinxiang, Henan 453003, China. [4]The Xinxiang Key Laboratory of Tumor Microenvironment and Immunotherapy, School of Laboratory Medicine, Xinxiang Medical University, Xinxiang, Henan 453003, China. [5]The Second Ward of Gynecological Tumor, Shaanxi Provincial Tumor Hospital, Xi'an, Shaanxi 710000, China. [6]The Key Laboratory of Shaanxi Province for Craniofacial Precision Medicine Research, Laboratory Center of Stomatology, Department of Orthodontics, College of Stomatology, Xi'an Jiaotong University, Xi'an, Shaanxi 710032, China. [7]The State Key Laboratory of Cancer Biology, Department of Immunology, Fourth Military Medical University, Xi'an, Shaanxi 710032, China. [8]These authors contributed equally: Xiang Zhang, Huilong Yin. ✉e-mail: ruizhang@fmmu.edu.cn

Among the more than one hundred RNA modifications, N6-methyladenosine (m6A) was identified in the 1970s[8]. M6A is the most abundant modification in eukaryotic mRNAs[9] and functions as an epitranscriptomic regulator of target mRNAs through multiple mechanisms, including regulating mRNA stability and translation efficiency[10]. As early as in 1990s, MT-A70, also known as METTL3 these years, was reported having methyltransferases activity of m6A[11]. However, approximately a decade ago, the identification of the first RNA demethylase, fat mass, and obesity-associated protein (FTO), revealed that N6-methyladenosine modification is reversible[12]. And consequently, a methyltransferase complex consisting mainly of methyltransferase-like 3 (METTL3), methyltransferase-like 14 (METTL14) and Wilms tumor-associated protein (WTAP) has been proven to act as an m6A methyltransferase ("writer") in mammalian cells[13,14] with METTL3 being the essential catalytic component of the complex[10,15–19]. Furthermore, a group of YT521-B homology (YTH) domain-containing family proteins (YTHDFs) have been identified as m6A "readers" that control mRNA fate by regulating pre-mRNA splicing, facilitating mRNA translation and promoting mRNA degradation[10,20–24]. Strikingly, Huang et al. and other groups demonstrated that insulin-like growth factor 2 (IGF2) mRNA-binding proteins 1, 2, and 3 (IGF2BP1/2/3) preferentially recognized m6A-modified mRNAs and promoted the stability (and likely the translation) of thousands of potential target mRNAs in an m6A-dependent manner, thereby affecting global gene expression output[25–28].

Importantly, with deeply understanding the biochemical process of N6-methyladenosine modification in the past decade, more studies have moved forward to explore the functional significance of m6A in various biological processes, including DNA damage repair[29], meiosis[30], circadian clock[31], and tumor immune surveillance[32]. Transcriptome-wide mapping of m6A modification has revealed cell type-specific methylation targets, suggesting that m6A regulates cell type-specific processes[33]. Liver is an important metabolic organ in the body, and maintaining its functional homeostasis is essential for health. It has been reported that inhibition of m6A methylation decreases the m6A abundance in PPaRα and increases the lifetime and expression of PPaRα mRNA, reducing lipid accumulation in hepatocytes[34]. In addition, m6A modification can also orchestrate sex-dimorphic metabolic traits in liver. Loss of m6A control in male livers increases hepatic triglyceride stores, leading to a more 'feminized' hepatic lipid composition[35]. All the evidence supports that dynamic modification of m6A is essential for the multiple physiological functions of liver tissues.

Here, we find a positive relationship between m6A levels and glycogen contents in the hepatocytes during mouse growth. Strikingly, hepatic glycogen is almost completely absent in mice with liver-specific Mettl3 deletion. Analysis of m6A-methylated RNA immunoprecipitation sequencing with qRT-PCR (MeRIP-qPCR) shows that METTL3 induces an increase in the m6A level in Gys2 mRNA and promotes IGF2BP2-mediated Gys2 mRNA stability. Importantly, reintroduction of exogenous GYS2 partially rescues glycogen storage in Mettl3-deficient hepatocytes in vivo. Furthermore, co-expression of Mettl3 and Gys2 is associated with liver glycogen storage in other mammals, such as Sprague-Dawley rats. Taken together, our study reveals that a METTL3-IGF2BP2-GYS2 axis potentially regulates the hepatic glycogen quantity.

## Results

### Mouse pups have both low glycogen storage and low mRNA m6A level in liver

Mice are usually weaned after 4 weeks of age and reach sexual maturity between 6 and 8 weeks. After 8 weeks of age, most mice have the ability to procreate. Here, we measured liver glycogen levels in C57BL/6 mice in a free diet at different ages and found that glycogen storage in the liver gradually increased with the growth of mice (Fig. 1a). Via

transmission electron microscopy (TEM), only a few glycogen molecules were observed in hepatocytes from 4-week-old mice; however, hepatocytes from 8-week-old mice contained an abundance of glycogen in the cytoplasm (Fig. 1b). Glycogen content assay confirmed this conclusion (Fig. 1c). Intriguingly, if we conjointly analyzed samples from different ages, the relative fold of m6A in livers' mRNAs had an obvious positive relationship with glycogen content (Fig. 1d).

### Genetic and pharmacological inhibition of Mettl3 simulates lack of glycogen in liver

To investigate why mRNA m6A levels increase with age, we analyzed an RNA-seq dataset with different-aged mouse livers (GSE58827). Mettl3, but not other key m6A machinery components, was found having a significant increase with age (Fig. 2a). And the elevation of METTL3 was also observed in protein level (Fig. 2b).

To demonstrate the detailed mechanism between Mettl3 and glycogen storage, firstly, we generated a hepatocyte-specific Mettl3 knockout model, albumin-Cre Mettl3^fl/fl (Supplementary Fig. 1a). As expected, the declines in METTL3 protein (Supplementary Fig. 1d) and in relative fold of global m6A mRNA (Supplementary Fig. 1c) were observed in cKO-mice's liver tissues. Intriguingly, the relative fold of m6A mRNA had no significant difference between WT (wild-type) and HET (heterozygous) mouse livers (Supplementary Fig. 1c), although they have different Mettl3 mRNA levels (Supplementary Fig. 1b). Glycogen content assay and periodic acid-Schiff (PAS) staining revealed that glycogen shortage in the liver was simulated in Mettl3-cKO mouse, but WT and HET mice had similar glycogen level in liver (Fig. 2c, d). Via transmission electron microscopy (TEM), almost no glycogen could be observed in hepatocytes of Mettl3-cKO mice (Fig. 2e). In addition, AQP8, a channel protein that was reported to have a close temporal and spatial correlation with glycogen accumulation in hepatocytes[36], was reduced in Mettl3-cKO mice (Supplementary Fig. 1e). Considering that the most important function of liver glycogen is blood glucose maintenance, we measured the serum glucose level in mice of different genotypes and found that it was greatly reduced in Mettl3-cKO mice (Fig. 2f). Furthermore, Mettl3-cKO mice also had worse performance than WT and HET mice in the forced swim test (Fig. 2g). In summary, hepatocyte-specific Mettl3-cKO could simulate glycogen shortage in liver, although Mettl3-HET and Mettl3-WT had no significant difference in glycogen and other associated phenotypes.

STM2457 is a highly efficient and selective catalytic inhibition on RNA methyltransferase METTL3. Eight-week-old male mice were sacrificed after treatment with vehicle or 50 mg/kg STM2457 each day for 5 days (Supplementary Fig. 2a). As excepted, relative fold of m6A mRNA was diminished in hepatocytes of STM2457-treated mice (Supplementary Fig. 2b). PAS staining (Supplementary Fig. 2c), transmission electron microscopy (Supplementary Fig. 2d) and glycogen content assay (Supplementary Fig. 2e) showed that STM2457-treated mice had much less glycogen in liver. Taken together, Mettl3-specific methyltransferase activity inhibition also simulate lack of glycogen in mouse liver.

### Identification of Gys2 mRNA as a key target of METTL3 in mouse liver

To investigate the role of m6A in liver glycogen storage, total RNA was isolated from samples of Mettl3-WT and Mettl3-cKO hepatocytes or liver tissues for m6A profiling by MeRIP-seq. Motif searching identified the consensus core "GGAC" motif within the m6A sites (Fig. 3a). The density of m6A peaks increased steadily along the transcript in the CDS and decreased along the 3'-UTR (Fig. 3b). Gene Ontology (GO) terms related to numerous metabolic processes were the most significantly enriched with genes exhibiting m6A peak loss (Supplementary Fig. 3a) and downregulated expression (Supplementary Fig. 3b) in Mettl3-cKO liver tissue. These results suggested that m6A modification has a considerable influence on metabolic pathways. Then, we obtained 26

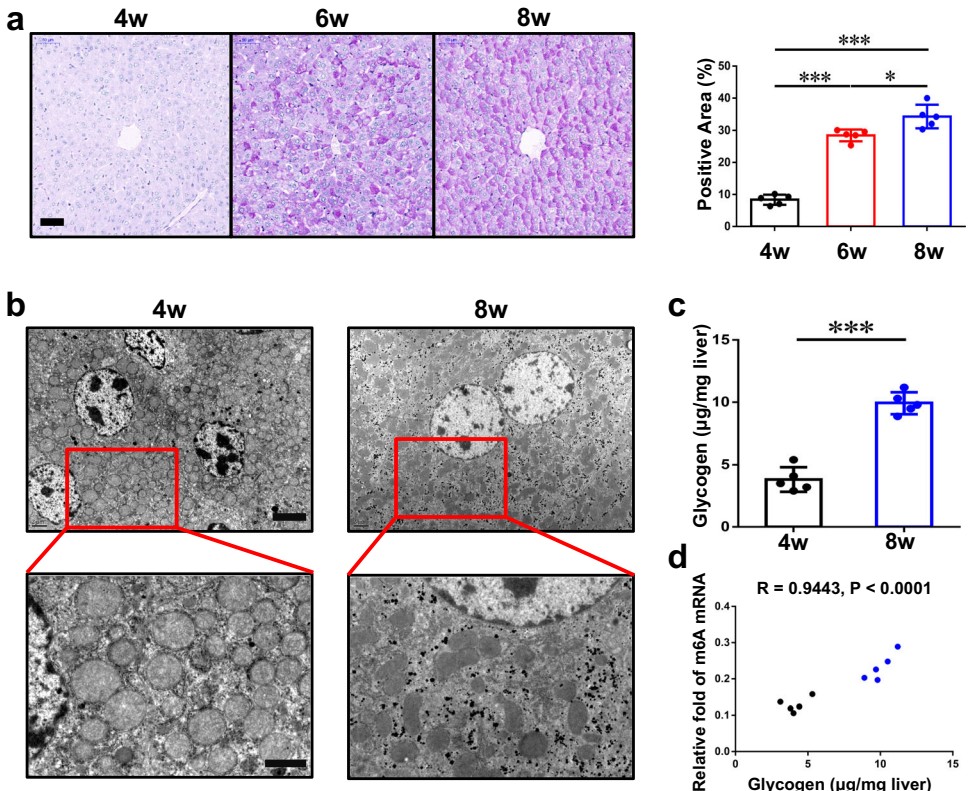

**Fig. 1 | Low content of mRNA m6A is related to a shortage of hepatic glycogen in mouse pups. a** PAS staining of wild-type mouse livers in different ages (4w, 4 weeks; 6w, 6 weeks; 8w, 8 weeks). The percentage of positive area is measured by Image J and shown on the right. Bar, 50 μm. $n = 5$ animals. Data are presented as mean values +/− SEM. *$p < 0.05$; ***$p < 0.001$ by one-way ANOVA. **b** Transmission electron microscope pictures of wild-type mouse livers in different ages. Bar: top, 4 μm; bottom, 2 μm. This experiment was repeated independently with similar results at least 3 times. **c** Hepatic glycogen content at different ages, as micrograms per milligram of tissue, $n = 5$ animals. Data are presented as mean values +/− SEM. Two-sided Student's t test was performed to determine a difference among groups. ***$p = 0.0006$. **d** The relationship between relative fold of m6A mRNA and hepatic glycogen content at different age samples. Black, 4w; blue, 8w. $n = 10$ animals. Linear regression was performed to determine a difference. Source data are provided as a Source data file.

candidate genes from one MeRIP-seq dataset (from 8-week-old mouse hepatocytes) and three independent RNA sequencing (RNA-seq) datasets (from 8-week-old mouse hepatocytes, male liver tissues and female liver tissues respectively). All 26 genes exhibited both m6A peak loss and a pattern of downregulated expression (Fig. 3c). Among these 26 genes was fatty acid synthase (Fasn) (Fig. 3c, Supplementary Fig. 3c), whose mRNA was reported to be a substrate of METTL3 in hepatocytes[37]. Intriguingly, this gene set contained glycogen synthase 2 (*Gys2*), the key enzyme coding gene of glycogenesis in liver (Fig. 3c, d). In addition, MeRIP-qPCR, performed in HET and cKO mouse hepatocytes, confirmed that *Gys2* mRNA had m6A modification in a METTL3-dependent manner (Fig. 3e). m6A MeRIP-Seq revealed that, like *Fasn* (Supplementary Fig. 3c), *Gys2* has highly enriched and specific m6A peaks among its coding sequence, but these peaks almost lost in *Mettl3*-cKO hepatocytes (Fig. 3d).

On expression influence of m6A modification, much lower *Gys2* level at both mRNA (Fig. 3f) and protein (Fig. 3g) were found in livers of *Mettl3*-cKO mice. Particularly worth mentioning is that pre-mRNA (nascent mRNA) of Gys2 had no significant different between *Mettl3*-HET and *Mettl3*-cKO mice (Supplementary Fig. 3d). It demonstrated that *Mettl3* depletion mainly influenced post transcription process of Gys2 expression. In addition, we tested mature and nascent Gys2 mRNA in STM2457 or vehicle-treated mouse livers, and found that mature but not nascent Gys2 mRNA had significant decrease under inhibition of METTL3 activity (Supplementary Fig. 3e, f). Finally, qRT-PCR assay illustrated that mature *Gys2* mRNA was higher in 8-week-old mouse liver than 4-week-old ones (Fig. 3h). MeRIP-qPCR and nascent mRNA qPCR assays demonstrated the different expression of Gys2 between

pup and adult mouse livers might be attributed to different methylation levels, but not transcription regulation (Supplementary Fig. 3g, h).

In conclusion, *Gys2*, the key enzyme of glycogenesis in hepatocytes, is a candidate target of METTL3-mediated m6A modification in liver. And loss of m6A modifications perishes *Gys2* mRNA expression in a post transcription manner.

## IGF2BP2 is involved in m6A mediated Gys2 mRNA stability

Different functions of m6A modification are reported to be mediated by different m6A binding proteins (i.e., readers), such as YTHDF1/2/3 and IGF2BP1/2/3[38]. As shown in Fig. 3f, the *Gys2* mRNA level was much lower in liver tissue in *Mettl3*-cKO mice than in control mice. To determine which reader proteins mediate the mRNA stability of *Gys2*, we independently depleted the six candidate readers in Hepa 1-6 (mouse hepatocellular carcinoma) cells and found that depletion of only IGF2BP2 reduced *Gys2* mRNA expression (Fig. 4a, Supplementary Fig. 4a). Furthermore, mRNA decay assays demonstrated that a reduction in IGF2BP2 expression appreciably promoted the degradation of *Gys2* mRNA (Fig. 4b). Finally, IGF2BP2 antibody-based RNA Immunoprecipitation (RIP) assays uncovered that the binding between IGF2BP2 protein and *Gys2* mRNA was m6A (METTL3) dependent (Fig. 4c, Supplementary Fig. 4d). However, Fasn, another target of Mettl3 in hepatocytes, had no significant difference in IGF2BP2 RIP-qPCR assay (Supplementary Fig. 4b). To sum up, IGF2BP2 could combine and stabilize Gys2 mRNA in a METTL3-dependent manner.

To investigate the binding site of IGF2BP2 and whether this binding is m6A dependent or not, we analyzed the sequences of different m6A peaks in hepatocytes' MeRIP-seq data by a SRAMP online

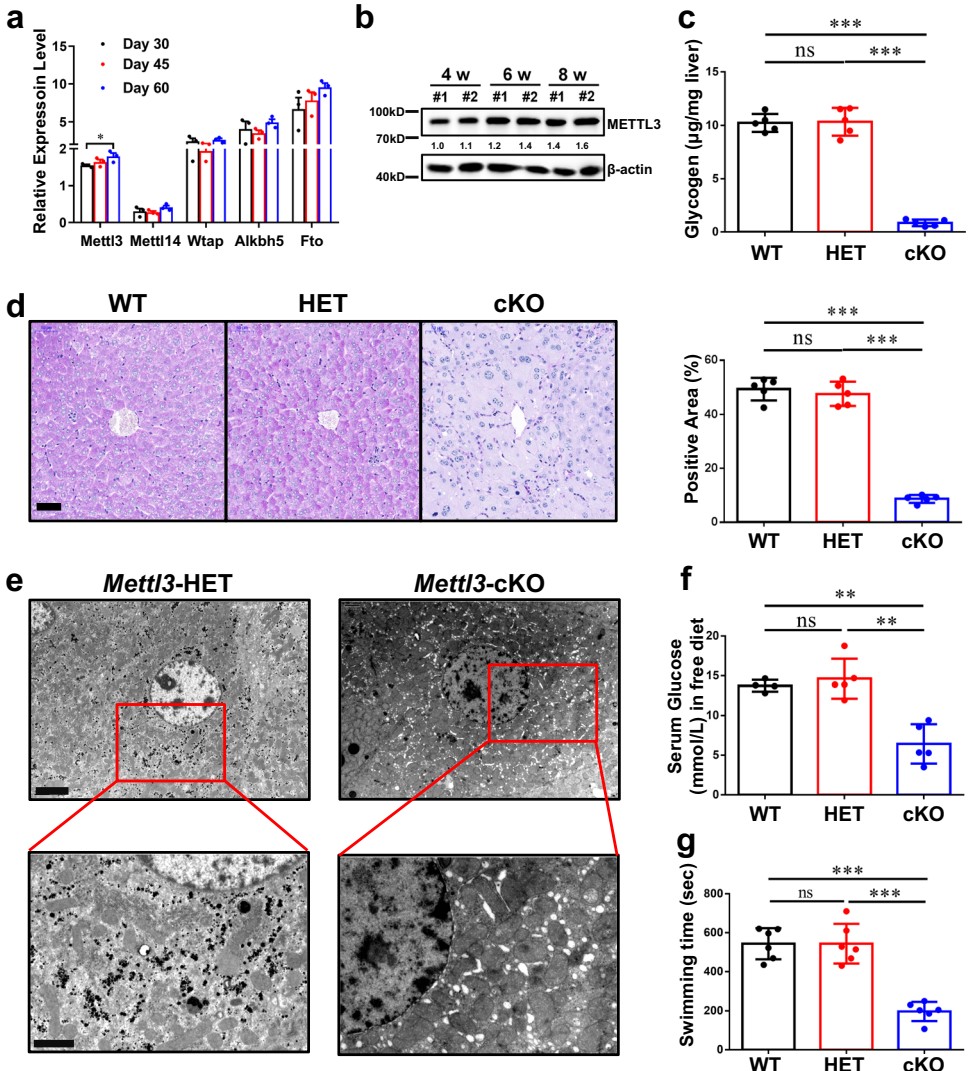

**Fig. 2 | *Mettl3* depletion simulates lack of glycogen in liver. a** Relative expression levels (FPKM) of five key genes in m6A writing or erasing from a mouse liver-associated GEO dataset (GSE58827). $n = 3$ independent samples. Data are presented as mean values +/− SEM. Two-way ANOVA was performed to determine a difference among columns within a gene. *$p < 0.05$. **b** Western blotting assay of *Mettl3* protein level in wild-type mouse livers from different ages. This experiment was repeated independently with similar results at least 3 times. **c** Hepatic glycogen content at different ages, as micrograms per milligram of tissue, $n = 5$ animals. Data are presented as mean values +/− SEM. One-way ANOVA was performed to determine a difference between each column. ns, not significant; ***$p < 0.001$. **d** PAS staining of livers in 8-week-old mice with different *Mettl3* genotypes. The percentage of positive area is measured by Image J and shown on the right. WT wild type, HET heterozygous, cKO conditional knockout (Albumin-cre). Bar, 50 μm. $n = 5$ animals.

Data are presented as mean values +/− SEM. One-way ANOVA was performed to determine a difference between each column. ns, not significant; ***$p < 0.001$. **e** Transmission electron microscope pictures of livers in 8-week-old mouse with indicated *Mettl3* genotypes. Bar: top, 4 μm; bottom, 2 μm. This experiment was repeated independently with similar results at least 3 times. **f** Serum glucose level of 8-week-old mice with different *Mettl3* genotypes. $n = 5$ animals. Data are presented as mean values +/− SEM. One-way ANOVA was performed to determine a difference between each column. ns, not significant; **$p < 0.01$. **g** Exhaustive swimming assay of 8-week-old mice with different *Mettl3* genotypes. $n = 6$ animals. Data are presented as mean values +/− SEM. One-way ANOVA was performed to determine a difference between each column. ns, not significant; ***$p < 0.001$. Source data are provided as a Source data file. See also Supplementary Figs. 1 and 2.

tool (http://www.cuilab.cn/sramp), and found two candidate sites (Fig. 4d). Then, we built two different "A to U" ("A to T" for DNA) mutant constructs of mouse Gys2 CDS (Fig. 4d). Based on the Flag tag in N terminal, we designed specific qPCR primers to detect exogenous mRNA of WT or mutant constructs (Supplementary Table 2). qRT-PCR (Fig. 4e) and Western blot assay (Fig. 4f) in Hepa 1-6 cells showed that +1172, not +2111, is the site that influences expression of Gys2 mRNA. And this conclusion was confirmed by dual luciferase report (DLR) assay (Fig. 4g). IGF2BP2-RIP-qPCR illustrated that mutation of +1172 site blocked the binding of IGF2BP2 protein to Gys2 mRNA (Fig. 4h, Supplementary Fig. 4c). Taken together, it depended on an m6A site (+1172 in CDS) that IGF2BP2 combined and stabilized Gys2 mRNA in mouse liver.

In conclusion, *Gys2* mRNA should be a bona fide substrate of METTL3 and that *Gys2* mRNA is stabilized by IGF2BP2 in an m6A-dependent manner.

## Reconstitution of GYS2 rescues liver glycogenesis in Mettl3-cKO mice

To verify the METTL3-IGF2BP2-GYS2 axis, we reconstituted GYS2 in the *Mettl3*-cKO mouse livers via adeno-associated virus (AAV)-mediated transduction. Compared to control AAV (AAV-luciferase, AAV-Luc), AAV-*Gys2* restored the GYS2 protein level in *Mettl3*-cKO mice to the baseline level (Fig. 5a, b). Glycogen content assay (Fig. 5c), PAS staining (Fig. 5d), and transmission electron microscopy (Fig. 5e)

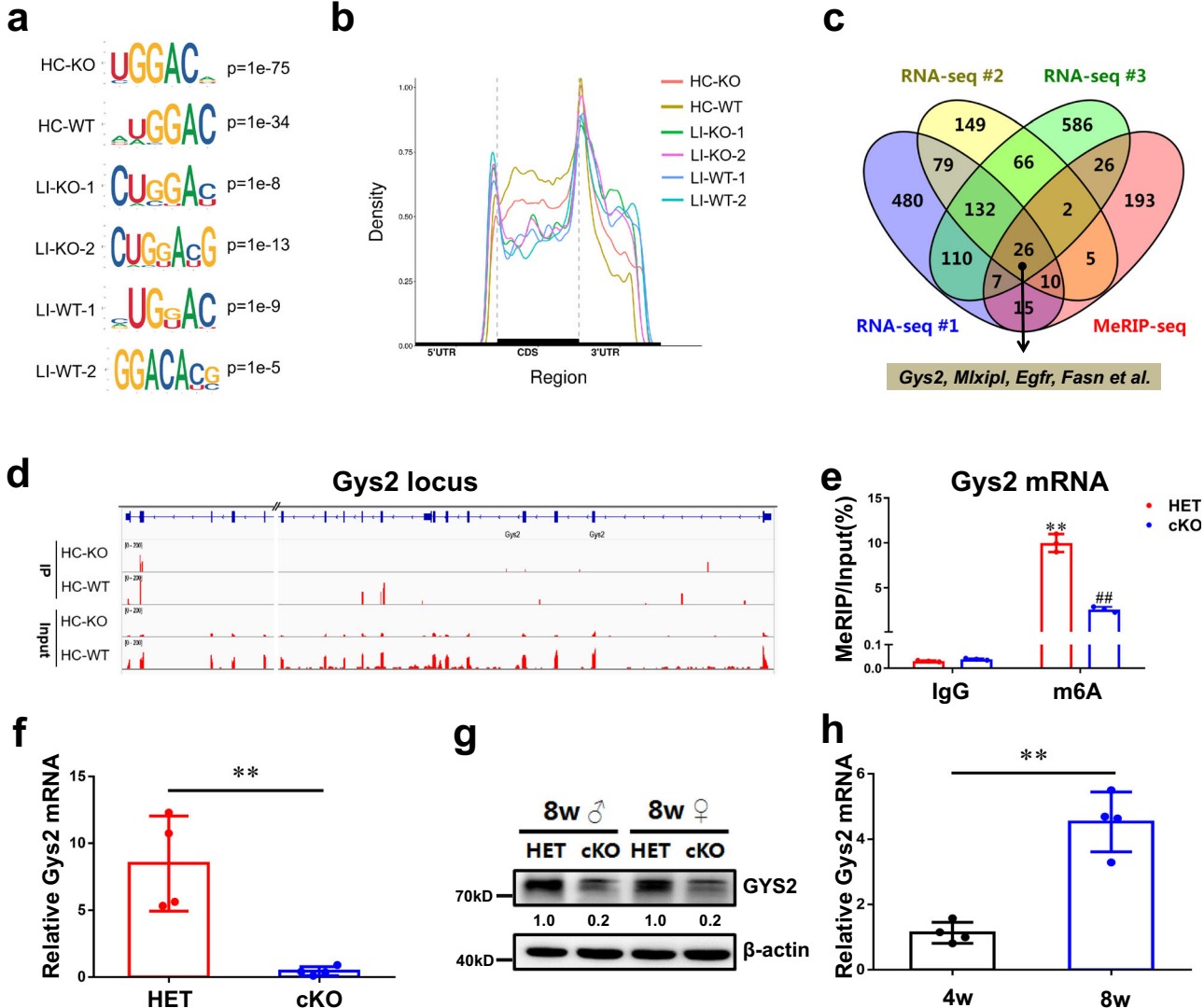

**Fig. 3 | Identification of *Gys2* mRNA by global N6-methyladenosine modification analysis as a key substrate of METTL3 in mouse liver. a** The most enriched sequence motif of m6A peaks in mRNAs from 8-week-old WT or cKO mouse hepatocytes (HC) or liver (LI) samples. **b** Distribution of m6A peaks along the 5′UTR, CDS, and 3′UTR regions of total mRNAs from 8-week-old mouse samples after normalized with length. **c** RNA-Seq and MeRIP-Seq identified downregulated and m6A-lost genes in *Mettl3*-cKO liver tissue or hepatocytes comparing to wild-type samples. All the samples were from 8-week-old mice. MeRIP-seq data were from hepatocytes of male mice; RNA-seq data were from hepatocytes and liver tissues of male mice and liver tissues of female mice, respectively. **d** m6A MeRIP-Seq revealed the location of specific m6A peak in *Gys2* locus in hepatocytes of wildtype or Mettl3-cKO mice. //, deletion of a long and uninformative intron for better exhibition. **e** m6A enrichment of *Gys2* mRNA in *mettl3*-HET or cKO hepatocytes by m6A-RIP-qPCR. $n = 3$ independent

experiments. Data are presented as mean values +/− SEM. Two-way ANOVA was performed to determine a difference among each two groups. **$p < 0.01$ comparing with IgG:HET; ##$p < 0.01$ comparing with m6A:HET. **f** qRT-PCR of *Gys2* mRNA levels (β-actin as reference) in 8-week-old mouse livers with indicated Mettl3 genotypes. $n = 4$ animals. Data are presented as mean values +/− SEM. Two-sided Student's t test was performed to determine a difference among groups. **$p = 0.004$. **g** Western blotting assay of GYS2 protein levels in 8-week-old mouse livers with indicated *Mettl3* genotypes. The ratio of GYS2 to β-actin were shown between two lanes. This experiment was repeated independently with similar results at least 3 times. ♂, male; ♀, female. **h** qRT-PCR of *Gys2* mRNA levels (β-actin as reference) in wildtype mouse livers with indicated ages. Two-sided Student's t test was performed to determine a difference among groups. **$p = 0.004$. $n = 4$ animals. Data are presented as mean values +/− SEM. Source data are provided as a Source data file.

demonstrated that reconstitution of GYS2 partially restored liver glycogen storage in *Mettl3*-cKO mice. Systemically, both the serum glucose level in *Mettl3*-cKO mice in a free diet (Fig. 5f) and the blood glucose level in *Mettl3*-cKO mice after prolonged fasting (Fig. 5g) increased under AAV-Gys2 treatment. In addition, *Mettl3*-cKO mice with AAV-Gys2 intervention performed better in the forced swim test than *Mettl3*-cKO mice with AAV-Luc intervention, although a performance gap remained between these mice and HET mice (Fig. 5h). In summary, reconstitution of GYS2 partially reversed *Mettl3*-cKO-associated glycogen deficiency and the related phenotypes in mice.

## Rats also have the METTL3-IGF2BP2-GYS2 axis in liver

Liver glycogen storage is very important for almost all mammals. To investigate whether the axis we found in mouse exists in other mammals, we test the samples from Sprague-Dawley (SD) rats. First, PAS staining (Fig. 6a) and glycogen content assay (Fig. 6b) revealed that glycogen storage in the liver is much lower in 4-week-old male rats than 8-week-old ones. It suggested that, similar to mouse, adult rats also have high level glycogen in liver comparing to pups. In addition, the relative fold of liver m6A mRNA had a similar pattern to that of glycogen storage at the indicated ages (Fig. 6c). Furthermore, qRT-PCR assay demonstrated that mRNA of *Mettl3* (Fig. 6d) and *Gys2* (Fig. 6e) in

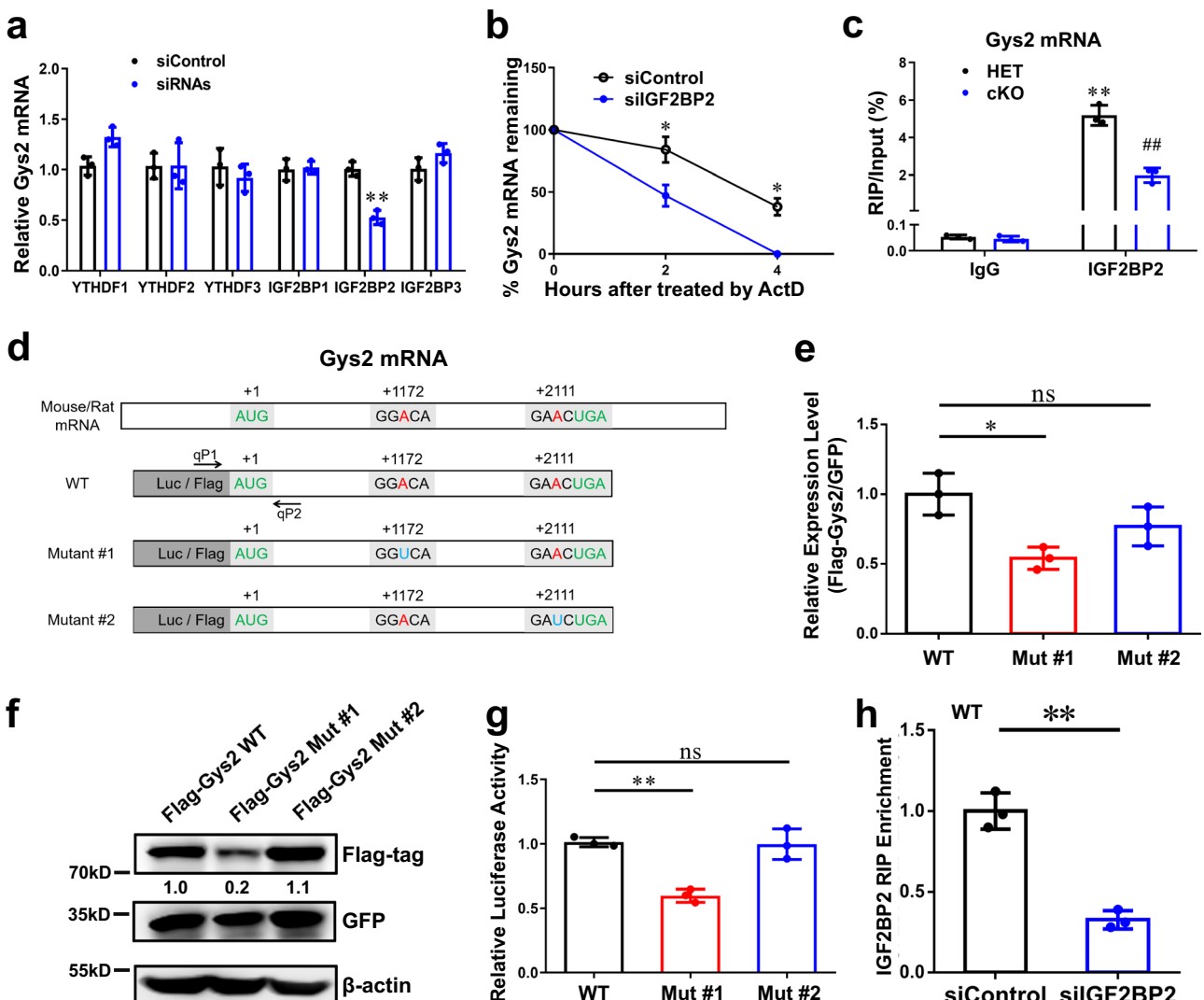

**Fig. 4 | N6-methyladenosine stabilizes *Gys2* mRNA in an IGF2BP2-dependent manner. a** qRT-PCR assay of relative *Gys2* mRNA level (β-actin as reference) in siControl and indicated siRNAs. $n = 3$ independent experiments. **$p < 0.01$ comparing with IGF2BP2:siControl. **b** qRT-PCR assay of *Gys2* mRNA stability (β-actin as reference) in siControl and siIGF2BP2 Hepa1-6 cells treated with actinomycin D (Act D) at the indicated times. $n = 3$ independent experiments. *$p = 0.022$ at hour 2; $p = 0.039$ at hour 4. **c** enrichment of *Gys2* mRNA in *mettl3*-HET or cKO hepatocytes by IGF2BP2-RIP-qPCR. $n = 3$ independent experiments. **$p < 0.01$ comparing with IgG:HET; ##$p < 0.01$ comparing with IGF2BP2:HET. **d** Two conserved candidate m6A sites were predicted by SRAMP (http://www.cuilab.cn/sramp), according to MeRIP-seq data. Mutant #1 and #2 were constructs with "A to U" mutation at different predicted sites. Luciferase (Luc) or Flag-tag were fused in N terminal of each construct. qP1 and qP2 were specific qRT-PCR primers to detect exogenous WT or Mutant Flag-Gys2 mRNAs. **e, f** Flag-Gys2-CDS WT or indicated Mut construct and GFP overexpression constructs were transfected

into Hepa1-6 cells for 48 h. Exogenous constructs expression levels were measured by qRT-PCR (**e**) and western blotting (**f**). GFP was reference in both assays. In figure **e**, $n = 3$ independent experiments. ns, not significant; *$p < 0.05$. The experiment in **f** was repeated independently with similar results at least 3 times. **g** pmirGLO Luc-Gys2-WT or indicated Mut construct was transfected into HEK-293T cells for 48 h. Relative luciferase activity (Fluc/Rluc) was tested by DLR assay. $n = 3$ independent experiments. ns, not significant; **$p < 0.01$. **h** enrichment of Flag-Gys2-CDS WT in siControl or siIGF2BP2 Hepa1-6 cells by IGF2BP2-RIP-qPCR. $n = 3$ independent experiments. **$p = 0.007$. For figure **a**–**c**, **e**, **g**, **h**, data are presented as mean values +/− SEM. For figure **a**, **c**, two-way ANOVA was performed to determine a difference among each two columns. For figure **b**, **h**, two-sided Student's t test was performed to determine a difference between siControl and siIGF2BP2. For figure **e**, **g**, one-way ANOVA was performed to determine a difference between each column. Source data are provided as a Source data file.

livers were high in 8-week-old rats and low in 4-week-old rats. Finally, we tested serum glucose of different age rats in free diet. As expected, 8-week-old rats have higher serum glucose than 4-week-old ones (Fig. 6f). Taken together, these results suggest that the METTL3-IGF2BP2-GYS2 axis we found in mouse may also exist in other mammals, such as rats.

## Discussion

Under physiological conditions, human blood contains only 4 to 6 grams of glucose, while the daily consumption of glucose for a normal person is approximately 160 grams[39]. Glycogen synthesis and

breakdown in the liver maintain the homeostasis of blood glucose between meals, and disorders of liver glycogen metabolism often cause severe outcomes and even death in some pediatric patients[1,2]. Previous articles and our data (Figs. 1a–c, 6a, b) revealed that the concentration of glycogen in the liver increases gradually from birth to adulthood in numerous mammals, such as mice, rats, rabbits, sheep, and rhesus macaques[3–5,36]. And the blood glucose level exhibits a similar pattern[6]. However, the detailed mechanism and biological meaning of this phenomenon remain unclear.

Among the more than one hundred RNA modifications, N6-methyladenosine (m6A) was identified in the 1970s[8]. M6A

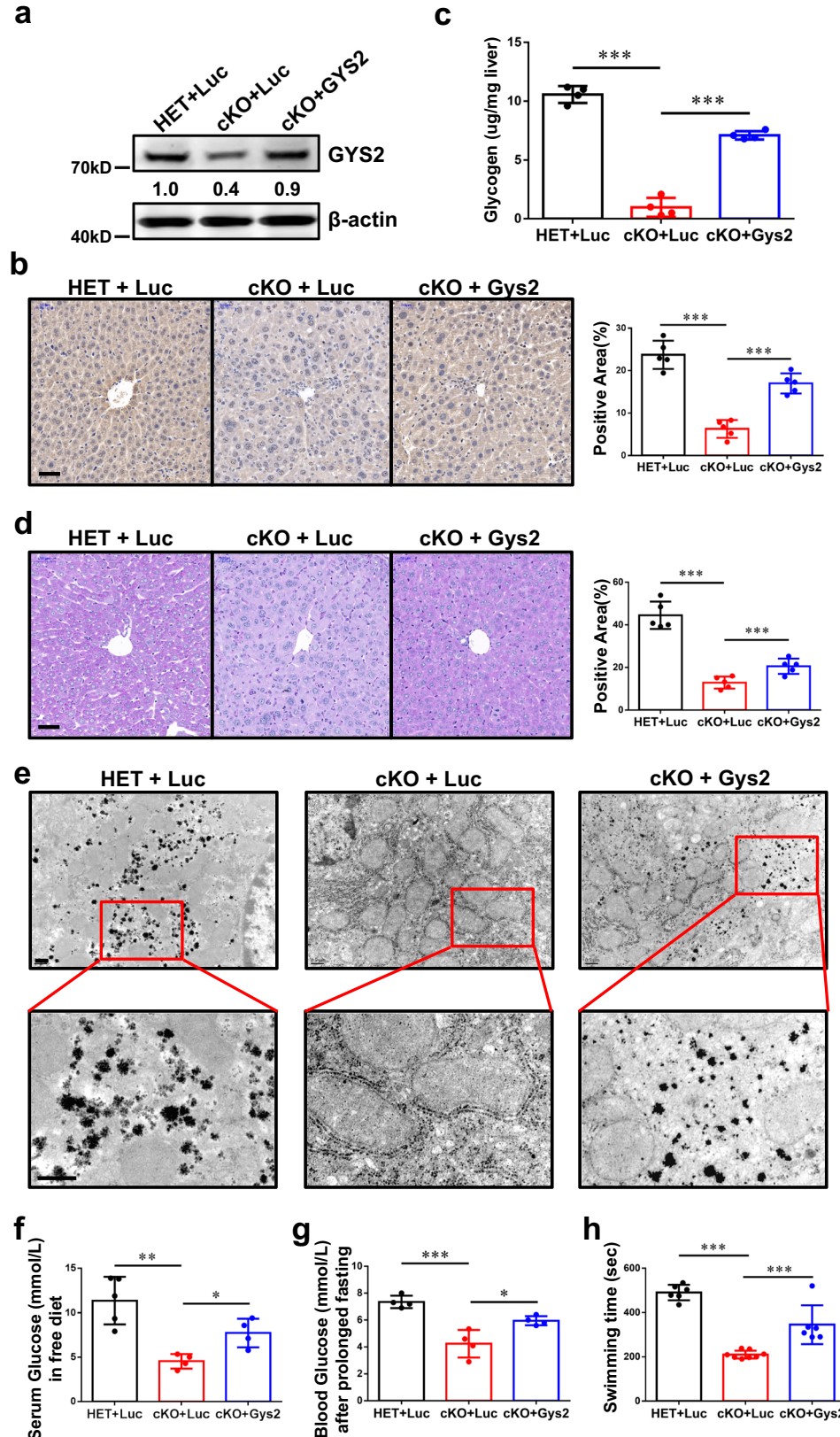

modification is the most abundant modification in eukaryotic mRNAs[9] and functions as an epitranscriptomic regulator of target mRNAs through multiple mechanisms, including enhancing stability and translation efficiency[10]. Intriguingly, we found that the relative fold of m6A mRNA had a very significant positive correlation with liver glycogen content among different ages (Fig. 1d). It

suggested that m6A modification of mRNA might be involved in glycogen metabolism.

Approximately a decade ago, an important report identified the first RNA demethylase (m6A "eraser"), fat mass and obesity-associated protein (FTO), revealing that RNA modification is reversible[12]. Then, ALKBH5 was confirmed as another m6A eraser in mammals[40]. On the

**Fig. 5 | Hepatic Gys2 overexpression alleviated glycogen shortage caused by Mettl3 knockout. a**, **b** Western blotting assay (**a**) and IHC staining (**b**) reveal the effect of reconstitution of GYS2. Luc, Luciferase. For figure **b**, bar = 50 μm. $n = 5$ animals. Data are presented as mean values +/− SEM. One-way ANOVA was performed to determine a difference between columns. ***$p < 0.001$. **c**–**e** Hepatic glycogen content assay (**c**) PAS staining (**d**) and transmission electron microscope pictures (**e**) of mouse livers with indicated genotypes and a luciferase (Luc) or Gys2-overexpressing adeno-associated virus (AAV). Bar: 50 μm (**d**), 0.5 μm (**e**). $n \geq 4$ animals. Data are presented as mean values +/− SEM. One-way ANOVA was performed to determine a difference between columns in figure (**c**) and (**d**).

***$p < 0.001$. **f** Serum glucose level of indicated groups of mice in free diet. $n \geq 4$ animals. Data are presented as mean values +/− SEM. One-way ANOVA was performed to determine a difference between columns. *$p < 0.05$; **$p < 0.01$. **g** Blood glucose level of indicated groups of mice after prolonged fasting. $n \geq 4$ animals. Data are presented as mean values +/− SEM. One-way ANOVA was performed to determine a difference between columns. *$p < 0.05$; ***$p < 0.001$. **h** Exhaustive swimming assay of indicated groups of mice. $n \geq 6$ animals. Data are presented as mean values +/− SEM. One-way ANOVA was performed to determine a difference between columns. ***$p < 0.001$. Source data are provided as a Source data file.

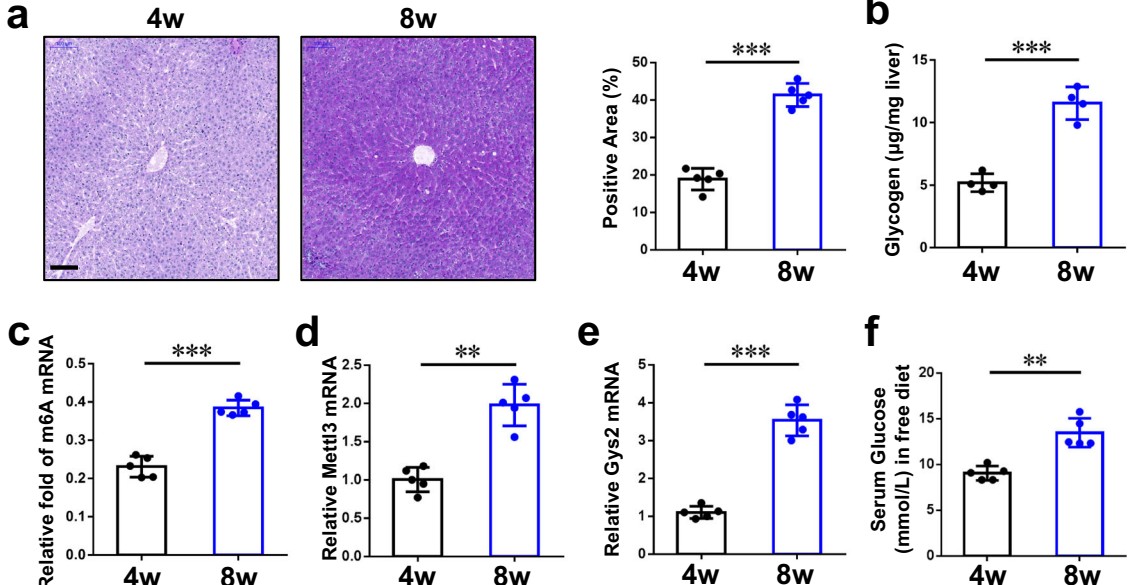

**Fig. 6 | The METTL3-IGF2BP2-GYS2 axis is related to liver glycogen storage in rats. a** PAS staining of wild-type rat livers in different ages. The percentage of positive area is measured by Image J and shown in the right. Bar, 100 μm. $n = 5$ animals. Data are presented as mean values +/− SEM. Two-sided Student's t test was performed to determine a difference among groups. ***$p < 0.001$. **b** Hepatic glycogen content assay of rat livers with indicated ages. $n = 4$ animals. Data are presented as mean values +/− SEM. Two-sided Student's t test was performed to determine a difference among groups. ***$p < 0.001$. **c** The levels of m6A mRNA were analyzed in rat livers of different ages. $n = 5$ animals. Data are presented as mean

values +/− SEM. Two-sided Student's t test was performed to determine a difference among groups. ***$p < 0.001$. **d**, **e** qRT-PCR assay of *Mettl3* (**d**) and *Gys2* (**e**) in rats' livers of different ages (β-actin as reference). $n = 5$ animals. Data are presented as mean values +/− SEM. Two-sided Student's t test was performed to determine a difference among groups. **$p = 0.002$ in (**d**); ***$p < 0.001$. **f** Serum glucose level of indicated ages of rats in normal diet. $n = 5$ animals. Data are presented as mean values +/− SEM. Two-sided Student's t test was performed to determine a difference among groups. **$p = 0.005$. Source data are provided as a Source data file.

other hand, m6A methyltransferases ("writers") in mammalian cells are complexes that contain various protein components. Methyltransferase-like 3 (METTL3) and methyltransferase-like 14 (METTL14) are still the core methyltransferases, although an increasing number of subunits have been verified[10,38]. An RNA-seq dataset with different-aged mouse livers (GSE58827) suggested that, *Mettl3*, but not other key components, had significant correlation with age (Fig. 2a). Then, western blotting assay also confirmed Mettl3's elevation during growth (Fig. 2b).

To investigate whether METTL3 plays an important role in liver glycogen storage, we generated a hepatocyte-specific *Mettl3* knockout model, albumin-Cre *Mettl3*fl/fl (Supplementary Fig. 1a–d). Notably, while our manuscript was in preparation, the findings we present here differ from the results of published data from Xu et al.[41]. Their results showed that albumin-Cre *Mettl3*fl/fl mice could only survival until 8 weeks old, and the expression of METTL3 decreased in wildtype mouse livers from 4-week-old to 8-week-old. We speculate that the reasons for this discrepancy may lie in the following: the *Mettl3* allele used in our strategy deleted the second and third exons in the Mettl3 gene (Supplementary Fig. 1a). Different strategies for the construction of *Mettl3*fl/fl mice may lead to inconsistent results. The diverse microbiome on account of different breeding environments also might influence the

homeostasis of livers[42,43]. This discrepancy suggests a complicated regulatory network behind m6A modification for controlling hepatocytes, which needs to be further clarified.

In our study, *Mettl3*-cKO mice had very low glycogen content (Fig. 2c–e, Supplementary Fig. 1e), although Mettl3-WT and Mettl3-HET mice had no significant different in glycogen storage (Fig. 2c, d). Furthermore, the serum glucose levels in *Mettl3*-cKO mice were much lower than other mice (Fig. 2f), and *Mettl3*-cKO mice also had worse performance in the forced swim test (Fig. 2g). Finally, liver glycogen defect was observed in a Mettl3 catalytic activity inhibition (STM2457 treatment) model (Supplementary Fig. 2a–e). Taken together, liver-specific depletion of Mettl3 could simulate glycogen shortage in liver, and this should be associated with loss of catalytic activity.

To identify substrates of METTL3 in the liver, we extracted total RNA from hepatocytes and liver tissues from wild-type (WT) and conditional knockout (cKO) mice. MeRIP-seq and input RNA-seq revealed that numerous metabolic processes were dysregulated (Supplementary Fig. 3a, b), suggesting that the principal differences in this model were established here. To enhance the representativeness of the screening, we used four different datasets to take the intersection. The meRIP-seq dataset contained methylation-downregulated genes in hepatocytes between Mettl3-cKO and Mettl3-WT mice (Fold

change >= 2000, *P* value <1e−15). The other three RNA-seq datasets came from downregulated genes in hepatocytes, male liver tissues and female liver tissues of 8-week-old Mettl3-cKO mice, respectively. With this stringent strategy, we got 26 candidate genes (Fig. 3c). Mlxipl, Egfr, Fasn and Gys2, which were reported to be associated with glycogen metabolism, were in this 26-genes set (Fig. 3c). It suggested that they should be bona fide candidates. Mlxipl is a deleted gene in Williams-Beuren syndrome, however, glycogen storage defect and hypoglycemia (two main phenotypes of *Mettl3*-cKO mice) are not symptoms of this syndrome[44]. According to IMPC (International Mouse Phenotyping Consortium), a famous mouse phenotype website (https://www.mousephenotype.org/), glycogen storage defect and hypoglycemia are not phenotype of Egfr-KO mice. Fasn, a key enzyme in fatty acid synthesis, is thought to be less associated with glycogen synthesis, although glycometabolism and lipid metabolism are connected. Finally, we focused on Gys2 which is liver glycogen synthase and catalyzes the rate-limiting step in the synthesis of glycogen. Furthermore, loss-of-function mutations of Gys2 cause type 0 Glycogen Storage Disease (GSD) in children, who have glycogen storage defect and hypoglycemia as main symptoms[1,7].

Similar to Fasn, a known substrate of METTL3 in liver, meRIP-seq peaks and MeRIP-qPCR revealed the enrichment of m6A modification in *Gys2* mRNA of Mettl3-WT, but much less in Mettl3-cKO (Fig. 3d, e, Supplementary Fig. 3c). As a result, loss of the enrichment of m6A peaks in *Gys2* mRNA accompanied by less *Gys2* mRNA and protein levels (Fig. 3f, g). Transcription regulation should be excluded, since no significant changes were observed in pre-mRNA of Gys2 between genotypes (Supplementary Fig. 3d). The results from METTL3 activity inhibition experiments also confirm these conclusions above (Supplementary Fig. 3e, f). What's more, the construct with mutation of the potential METTL3 binding site in the *Gys2* CDS retained a much lower expression level than the WT construct (Fig. 4e–g). Finally, we observed 8-week-old mice had more mature *Gys2* mRNA than 4-week-old mice (Fig. 3h), but no significant difference in pre-mRNA of *Gys2* between ages (Supplementary Fig. 3h). And meRIP-qPCR assay demonstrated much more enrichment of m6A modification in 8-week-old mouse hepatocytes (Supplementary Fig. 3g). Taken together, these results indicated that *Gys2* mRNA should be a bona fide substrate of METTL3 in the mouse liver, and played an important role in glycogen regulation between pups and adults.

Like other epigenetic modifications, the functions of RNA m6A modification are carried out by RNA-binding proteins called m6A readers. Previously, a group of YT521-B homology (YTH) domain-containing family proteins (YTHDFs) were identified as m6A readers that control mRNA fate by regulating pre-mRNA splicing, facilitating mRNA translation and promoting mRNA decay[10,20–24]. Later, an increasing number of readers were identified[38]. Among these readers, insulin-like growth factor 2 (IGF2) mRNA-binding proteins 1, 2, and 3 (IGF2BP1/2/3) were reported by Huang et al. to preferentially recognize m6A-modified mRNAs and promote the stability (and probably also the translation) of thousands of potential mRNA targets in an m6A-dependent manner, thereby affecting global gene expression output[25]. In our study, a striking effect of *Mettl3* depletion was a substantially diminished *Gys2* mRNA level (Fig. 3f). Thus, we focused on readers that can stabilize target mRNAs. IGF2BP2, but not the other candidate proteins (IGF2BP1/3, YTHDF1/2/3), was essential to *Gys2* mRNA expression in Hepa 1-6 cells (Fig. 4a, Supplementary Fig. 4a). In addition, knockdown of IGF2BP2 in Hepa 1-6 cells could significantly shorten half-life of *Gys2* mRNA (Fig. 4b). Then, IGF2BP2-meRIP-qPCR assay confirmed binding of IGF2BP2 and *Gys2* mRNA, and this binding was METTL3 and m6A site-dependent (Fig. 4c, h, Supplementary Fig. 4c, d). In summary, IGF2BP2 emerged as a reader protein in METTL3-mediated Gys2 mRNA stability.

Logically, if the METTL3-IGF2BP2-GYS2 axis exists in cells, reconstitution of GYS2 would rescue *Mettl3*-cKO-associated hepatic glycogen deficiency. Here, reconstitution of GYS2 in hepatocytes of *Mettl3*-cKO mice by adeno-associated virus (AAV)-mediated transduction partially enhanced the accumulation of glycogen in the liver (Fig. 5c−e), improved serum glucose in free diet-fed mice (Fig. 5f), and increased blood glucose in mice after prolonged fasting (Fig. 5g). Moreover, the performance of *Mettl3*-cKO mice in the forced swim test was enhanced under AAV-GYS2 intervention (Fig. 5h). In conclusion, reconstitution of GYS2 partially rescued the unfavorable phenotypes associated with *Mettl3*-cKO in mice.

So far, our study demonstrated a METTL3-IGF2BP2-GYS2 axis that controls glycogenesis in liver. However, we could not completely exclude other factors that may also facilitate METTL3-associated glycogen storage, since liver contains various effectors to regulate glycogenesis. Life is complicated, and it usually have diverse mechanisms to regulate important phenotypes. For example, on core mechanism of m6A-regulated macrophage activation, we found SPRED2 was target of METTL3[24], however, other team found IRAKM could be another key gene in this process[45].

Does the METTL3-IGF2BP2-GYS2 axis exist in other mammals, such as rats? We investigated samples from male rats in different ages, and found that adult rats indeed have more glycogen storage than pups (Fig. 6a, b). As expected, the relative fold of m6A mRNA in livers have similar patterns to glycogen storage in indicated ages (Fig. 6c). Furthermore, *Mettl3* and *Gys2* also have much higher expression in livers of adult rats than pups (Fig. 6d, e). In addition, serum glucose of 8-week-old rats is higher than that of 4-week-old ones (Fig. 6f). Taken together, the phenomenon and mechanism, we found in mice, could also exist in other mammals, such as rats.

In summary, we revealed a METTL3-IGF2BP2-GYS2 axis that controlled glycogenesis in mammalian liver. On the one hand, this axis limited liver glycogen storage in mammalian pups to a very low level. As a result, a shortage of glycogen in the liver could improve the safety of animal pups in the wild by necessitating a short interval between feedings by their parents. On the other hand, this axis increases liver glycogen storage in adult mammals. Interestingly, under this condition, the abundance of glycogen in the liver could also improve the safety and facilitate the survival of adult mammals by allowing a long interval between meals in the wild (Fig. 7).

## Methods
### Mice/rats
Methyltransferase-like 3 (*Mettl3*)-floxed mice were generated by Beijing Biocytogen Co., Ltd. Albumin (Alb)-Cre mice purchased from the Jackson Laboratory were crossed with Mettl3-floxed mice to generate Alb-cre *Mettl3*[−/−] (wild-type, WT), Alb-cre *Mettl3*[fl/-] (HET) and Alb-cre *Mettl3*[fl/fl] (KO) mice. The PCR primers used for genotyping of each mouse strain are listed in Supplementary Table 1. All mice were on the C57BL/6 genetic background and were housed in individual cages. Littermate WT or HET mice were used as the control animals in each experiment. Experimental Sprague-Dawley (SD) rats were purchased from Lab Animal Center of the Fourth Military Medical University. Without special instructions, experimental mice/rats were bred and maintained under specific pathogen-free conditions, fed standard laboratory chow, and kept on a 12 h light/dark cycle (zeitgeber time[ZT]0-ZT24). All animal experiments were approved by the Animal Experiment Administration Committee of the Fourth Military Medical University.

### Animal studies

**Hepatic glycogen content.** Each 10 mg liver tissues were homogenized with 200 μL 30% KOH on ice and the homogenate was boiled for 10 min to inactivate enzymes. The boiled samples were centrifuged at $13,400 \times g$ for 10 min at 4 °C to remove insoluble materials and the supernatant was ready for the assay using a Glycogen Assay Kit II (colorimetric) (Abcam, ab169558), and the results were normalized to the weight of initial tissues.

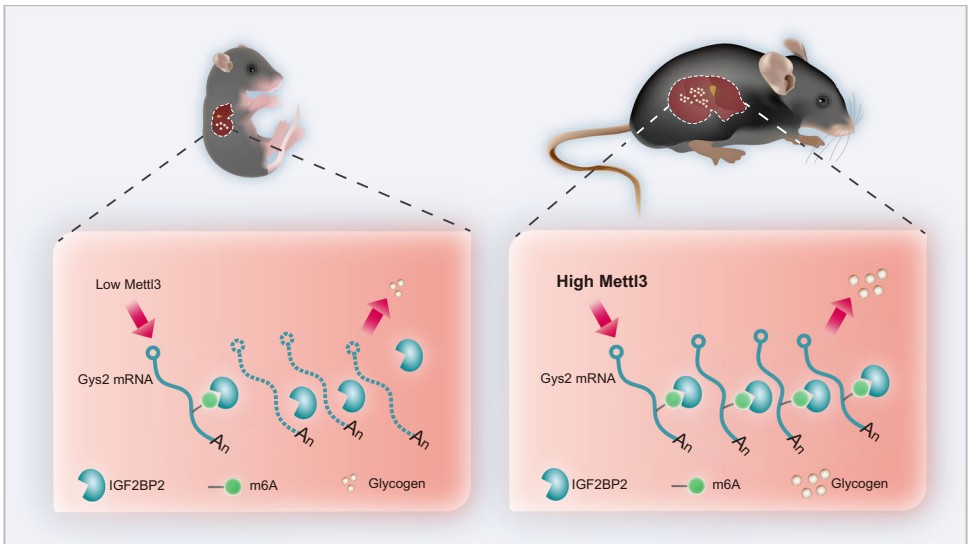

**Fig. 7 | Model for m6A regulation of Gys2 mRNA in the liver.** METTL3-IGF2BP2-GYS2 axis controls glycogenesis in mammalian liver. On the one hand, this axis limited liver glycogen storage in mammalian pups to a very low level. On the other hand, this axis increases liver glycogen storage in adult mammals.

**Serum glucose detection (mice/rats).** Normal diet-fed mice with the indicated genotypes (8-week-old male mice, $n \geq 5$) were sacrificed during ZT9-11. For rats, normal diet-fed ones with the indicated ages (male, $n \geq 5$) were sacrificed during ZT9-11. Total blood was collected and centrifuged (1500 g/min, 5 min) to isolate serum. The serum glucose level was measured by Servicebio, Inc.

**Blood glucose detection.** Mice with the indicated genotypes (8-week-old male mice, $n \geq 5$) fasted for 12–24 h. Blood glucose was measured using blood collected from mouse tail veins under sedation during ZT9-11.

**Forced swim test.** Mice with the indicated genotypes (8-week-old male mice, $n \geq 5$) were tested with a method based on a previous report[46].

**Mettl3 inhibitor (STM2457) assay.** Eight-week-old male C57BL/6 mice were i.p. treated daily for 5 days with either vehicle or 50 mg/kg STM2457 (Selleck™, S9870). Then, mice were sacrificed to detect glycogen or others.

**Adeno-associated virus (AAV)-mediated reconstitution of liver-specific glycogen synthase (Gys2).** AAVs containing luciferase (Luc) or Gys2 constructs were injected i.v. into mice with the indicated genotypes (8-week-old male mice, $n \geq 6$). Three weeks after injection, mice were evaluated by the indicated assays.

**Glycogen periodic acid-Schiff (PAS) staining and histological detection**
Qualitative assessment of liver tissue glycogen was performed using periodic acid-Schiff (PAS) staining. Mice/rats with the indicated genotypes or ages ($n \geq 5$) were sacrificed. The livers were rapidly harvested, and slices of the left and medial lobes were obtained and post fixed overnight with 4% paraformaldehyde in phosphate-buffered saline. Liver tissues were embedded in paraffin, sectioned (4 μm), and stained with PAS following the protocol of Vesselinovitch et al.[47]. Immunohistochemical assays were performed on FFPE sections as reported previously[48]. Antibodies against Gys2 (1:100, 22371-1-AP, Proteintech®) were used. Staining intensities were measured using Image J software (National Institutes of Health, Bethesda, MD, USA).

**Transmission electron microscopy (TEM)**
Liver samples were obtained from the left lobe and were fixed with 2.5% glutaraldehyde (pH = 7.2) and 1% osmium tetroxide and processed as described previously[49]. The slices were examined with a transmission electron microscope (JEM-1230, JEOL Ltd., Tokyo, Japan). Images were acquired by a technician blinded to the treatment.

**Isolation of hepatocytes**
Mice were perfused with 15 mL of prewarmed collagenase type I (0.1% w/v, Gibco) through the portal vein for 15 min. Livers were then removed and minced, and hepatocytes were pelleted by centrifugation at $50 \times g$ for 3 min three times. The purity of hepatocytes exceeded 90%.

**Cell culture**
Hepa 1-6 (mouse hepatocellular carcinoma) and HEK-293T cells were maintained in Dulbecco's modified Eagle's medium (DMEM; Gibco BRL, USA) supplemented with 10% fetal calf serum (FCS; Gibco BRL, USA) and antibiotics (complete medium). Cells were incubated in a humidified atmosphere of 5% $CO_2$ in air at 37 °C.

**AAV vector production**
Recombinant AAV vectors were produced by a standard triple-transfection calcium phosphate precipitation method using AAV-293 cells. The production plasmids (AAV-MCS, AAV-DJ, and pHelper) were from the AAV-DJ Helper Free Expression System (CELL BIOLABS, VPK-410-DJ), and the AAV-Luc and AAV-Gys2 constructs were generated from AAV-MCS. Purification was performed with clarified AAV-293 cell lysates using a ViraBind™ AAV Purification Mega Kit (CELL BIOLABS, VPK-141). Viral genome (vg) titers were determined with a QuickTiter™ AAV Quantitation Kit (CELL BIOLABS, VPK-145).

**Quantitative real-time PCR (qRT-PCR)**
Total RNA was isolated from cells using TRIzol (Invitrogen). For qRT-PCR analysis of nascent and mature mRNAs, first-strand cDNA was synthesized using a Prime Script qRT-PCR Kit (Takara). The amplification signal of qPCR data was acquired by Bio-Rad CFX Manager 3.1. The expression levels of the target genes were determined by amplification with specific primers. The primers used are listed in Supplementary Table 2.

### N6-methyladenosine (m6A)-sequencing (m6A-seq) and quantification of mRNA methylation by m6A-methylated RNA immunoprecipitation sequencing with RT-qPCR (MeRIP-qPCR)

Total RNA was isolated from cells using TRIzol (Invitrogen). MeRIP-Seq was performed by Cloudseq Biotech Inc. (Shanghai, China) according to the published procedure with slight modifications. Briefly, m6A RNA immunoprecipitation was performed with the GenSeq™ m6A RNA IP Kit (GenSeq Inc., China) by following the manufacturer's instructions. Both the input sample without immunoprecipitation and the m6A IP samples were used for RNA-seq library generation with NEBNext® Ultra II Directional RNA Library Prep Kit (New England Biolabs, Inc., USA). The library quality was evaluated with BioAnalyzer 2100 system (Agilent Technologies, Inc., USA). Library sequencing was performed on an illumina Hiseq instrument with 150 bp paired-end reads.

Briefly, Paired-end reads were harvested from Illumina HiSeq 4000 sequencer, and were quality controlled by Q30. After 3' adapter-trimming and low-quality reads removing by cutadapt software (v1.9.3). First, clean reads of all libraries were aligned to the reference genome (UCSC MM10) by Hisat2 software (v2.0.4). Methylated sites on RNAs (peaks) were identified by MACS software. Differentially methylated sites were identified by diffReps. These peaks identified by both softwares overlapping with exons of mRNA were figured out and chosen by home-made scripts. GO and Pathway enrichment analysis were performed by the differentially methylated protein-coding genes.

For MeRIP-qPCR, poly(A)$^+$ mRNA was isolated using a Dynabeads™ mRNA Direct Purification Kit (61006, Invitrogen). Purified RNA (2 μg) was used for enrichment of m6A-containing mRNA with a Magna MeRIP™ m⁶A Kit (17-10499-1, Millipore, 5 μg antibody per sample), and the RNA was purified according to the manufacturer's protocol. The resulting final product was used for qRT-PCR. The primers used to amplify mouse mature Gys2 mRNA are listed in Supplementary Table 2.

### Relative quantification of mRNA m6A methylation

mRNA was isolated using the Dynabeads™ mRNA Direct Purification Kit (61006, Invitrogen). Methylation of purified mRNA was quantified using an EpiQuik m6A RNA Methylation Quantification Kit (P-9005, EpiGentek) according to the manufacturer's protocol.

### RNA immunoprecipitation-qPCR (RIP-qPCR)

This procedure was performed according to a previously published report[24]. Hepa 1-6 cells or mettl3-HET/cKO hepatocytes were washed twice with PBS and lysed in lysis buffer (150 mM KCl, 10 mM HEPES (pH 7.6), 2 mM EDTA, 0.5% NP-40, 0.5 mM dithiothreitol (DTT), 1:100 protease inhibitor cocktail, 400 U/ml RNase inhibitor). The cell lysates were centrifuged. A 50-μl aliquot of the cell lysate was saved as input, and the remaining sample was incubated with 20 μl of protein A beads previously bound to 5 μg IgG or an anti-insulin-like growth factor 2 mRNA-binding protein 2 (IGF2BP2) antibody (Proteintech™) for 4 h at 4 °C. The beads were washed 2 times with wash buffer (50 mM Tris, 200 mM NaCl, 2 mM EDTA, 0.05% NP40, 0.5 mM DTT, RNase inhibitor). RNA was eluted from the beads with 50 μl of RLT buffer and purified with RNeasy columns (217004, QIAGEN). RNA was eluted in 100 μl of RNase-free water and reverse transcribed into cDNA using a Prime Script qRT-PCR Kit (Takara) according to the manufacturer's instructions. The fold enrichment was determined by qRT-PCR. The primers used for amplifying mouse mature Gys2 and Fasn mRNA are listed in Supplementary Table 2.

### mRNA stability analysis

To evaluate mRNA stability, Hepa 1-6 cells, with normal or knockdown IGF2BP2, were treated with actinomycin D (Sigma) at a final concentration of 5 μg/ml for 0, 2, or 4 h. The cells were collected, and RNA samples were extracted for reverse transcription. The mRNA transcript levels of interest were determined by qRT-PCR.

### Dual-luciferase reporter assay

Transfection was performed using Lipofectamine 2000 (Invitrogen). pmirGLO Fluc-Gys2 WT or indicated Mutant construct was transfected into HEK-293T cells for 48 h. Cell extracts were prepared and luciferase activity was measured using the Dual Luciferase Reporter Assay System (Promega, Madison, WI, USA). The relative firefly luciferase (Fluc) activity was normalized with its respective Renilla luciferase (Rluc) activity.

### Western blot analysis

Cells were harvested and lysed with RIPA buffer. The protein concentration was determined using a BCA kit. Samples were separated on 10% SDS-PAGE gels and blotted onto nitrocellulose membranes (Millipore). Membranes were incubated at 4 °C overnight with primary antibodies at the following concentrations: anti-GYS2 (1:2000, 22371-1-AP, Proteintech®), anti-IGF2BP2 (1:1000, 11601-1-AP, Proteintech®), anti-METTL3 (1:1000, ab195352, EPR18810, Abcam), anti-Flag-tag (1:1000, F1804, Sigma), anti-GFP (1:1000, 50430-2-AP, Proteintech®), and anti-β-actin (1:4000, Sigma). Membranes were then washed three times with TBST and incubated with HRP-conjugated anti-mouse IgG (1:10,000, 7076, Cell Signaling Technology) or anti-rabbit IgG (1:10,000, 7074, Cell Signaling Technology) diluted in TBST containing 1% non-fat milk at room temperature for 1 h. After a final wash with TBST, the membranes were developed with ECL reagents and visualized using a Tanon 5500 imaging system. The ratio of the expression of the indicated molecule to that of β-actin was determined using ImageJ software (National Institutes of Health, Bethesda, MD, USA).

### Bioinformatic analysis

Figure 3a, b: Motif analysis was performed using HOMER(Version 4.10)[50] to search motifs in each set of m6A peaks. Metagene profiles were generated as described[51]. Peak density plot was visualized by R package Trumpet (https://github.com/skyhorsetomoon/Trumpet)[52]. Supplementary Fig. 3a, b: Gene Ontology (GO) biological process (BP) enrichment was analyzed by CloudSeq Biotech; $p < 0.05$ was considered to indicate significant enrichment.

A different-age mouse liver RNA-seq dataset with the GEO series accession number GSE58827 was download from NCBI's Gene Expression Omnibus.

### Statistical analysis

Data were analyzed using GraphPad Prism (Version 6.02). Linear regression was performed to judge the relationship between relative fold of m6A mRNA and hepatic glycogen content at different age samples in Fig. 1d. Kolmogorov-Smirnov test was performed to determine significance in Supplementary Fig. 3a, b. Two-tailed Student's t-test, one-way ANOVA, and two-way ANOVA were performed with $P < 0.05$ considered significant. Exact information was in legends of indicated figures.

### Reporting summary

Further information on research design is available in the Nature Portfolio Reporting Summary linked to this article.

## Data availability

Raw data for RNA-sequencing and m6A-RIP sequencing have been deposited in NCBI's Gene Expression Omnibus and are accessible through GEO Series accession number GSE207566. Sequencing reads were mapped to themousemm10 genome. A different-age mouse liver RNA-seq dataset with GEO Series accession number GSE58827 was obtained from NCBI's GEO. All other data analyzed or generated during this study are included in this published article and its supplementary information files. Source data are provided with this paper.

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

## Acknowledgements

This work was supported by grants from the National Natural Science Foundation of China (81630069, 81421003, and 31771439 to A.G.Y.; 81773262, 81572763 and 82173046 to R.Z.; 82173162 to X.Z.; 31801128 to H.L.Y.); the Grant of Youth Training Project of PLA Medical Science and Technology, China (18QNP018 to X.Z.); the Key Research and Development Project of Shaanxi Province (2020SF-137 to X.Z.); and the Fund of State Key Laboratory of Cancer Biology (CBSKL2015Z15 to X.Z.).

## Author contributions

Conceptualization: Xiang.Z. and R.Z.; methodology: Xiang.Z., H.Y., and Xiao.Z.; investigation: Xiang.Z., H.Y., H.Z., Y.P., D.L., Y.Y., J.Z., X.J., S.C., Y.L., and Xiao.Z.; writing—original draft: Xiang.Z. and R.Z.; writing—review & editing: Xiang.Z. and R.Z.; funding acquisition: Xiang.Z., R.Z., H.Y., and A.Y.; resources: Xiang.Z. and R.Z.; Supervision: R.Z.

## Competing interests

The authors declare no competing interests.
