## [Peer Review File · Nature Communications]

N6-methyladenosine modification governs liver glycogenesis by stabilizing the glycogen synthase 2 mRNAReviewers' comments:

Reviewer #1 (Remarks to the Author):

In this manuscript, Zhang et al. find that that m6A levels in the livers of mice and rats increase with age and that m6A regulates the expression of Gys2 mRNA through the m6A reader protein IGF2BP2. Overall, this works needs to more clearly demonstrate the relationship between m6A, m6A reader proteins and Gys2 expression as stated in the points below:

Major points:

- 1) Liver mRNA m6A levels increase with age of animals. Why is this? A simple explanation could be an increase in the expression of methyltransferase subunits like METTL3, METTL14 or WTAP. It could also be due to decreased expression of demethylases like FTO or ALKBH5. The protein expression of these key m6A machinery components should be analyzed.
- 2) The Methods section lacks details about how the MeRIP-seq data was analyzed. Which programs were used? This is key information that is required to ensure that a robust pipeline was used. Were the peak heights normalized to input to account for changes in expression?
- 3) The authors should show tracks from each replicate for the Gys2 locus in Fig 3g. This will help demonstrate the consistency of the phenotype between replicates. Further, the authors should always show the input tracks as well to account for differences in expression.
- 4) While the IGF2BP family have been described as m6A readers, their exact role in binding m6A is still uncertain. Despite mentioning the critical m6A readers YTHDF1-3, the authors do not investigate whether these have any effect on Gys2 expression. This choice seems rather arbitrary. Therefore, the authors should test whether depletion of YTHDF members influences Gys2 as well.
- 5) The m6A mutants in Figures 4e and 4f are in the CDS. The authors should confirm that these are synonymous mutations and that the amino acid sequence of the proteins is not altered as this is currently not described. This is key as changes in the amino acid sequence may influence protein stability.
- 6) The authors test whether IGF2BP2 binds Gys2 in Figure 4d by depleting the RBP and testing interaction with Gys2. This, of course, indicates that IGF2BP2 depletion reduces the amount of pulled down Gys2 mRNA. However, this is obviously because of reduced IGF2BP2 in the depleted cells in the first place! A critical experiment would be if IGF2BP2 binding to Gys2 is reduced in context of METTL3 depletion or in the cKO. This would demonstrate whether or not IGF2BP2 interaction with Gys2 mRNA is dependent on m6A modification. Immunoblots of Input and IP fractions should be shown for RIP experiments to accurately draw conclusions.
- 7) The authors demonstrate that depletion of IGF2BP2 slightly reduces the stability of Gys2 mRNA. However, is this effect dependent on m6A? The authors should perform similar experiments with the m6A-mutant generated in figure 4e/f in the context of IGF2BP2 depletion (assuming they are synonymous mutants).
- 8) Given that the difference in Gys2 mRNA stability is slight, it is possible that affecting m6A pathways indirectly influences Gys2 mRNA by regulating a transcription factor instead. 4sU pulse chase experiments will determine differences in Gys2 transcription following METTL3 or IGF2BP2 depletion and can also provide insights into differential stability.
- 9) In Figure 4f, what are the RNA levels of the FLAG-tagged WT or m6A-mutant Gys2 constructs? This would help to confirm whether differences in RNA stability/expression are m6A-dependent.

10) Recently, several small molecule inhibitors of the methyltransferase complex have been discovered, including the older DAA and the newer STM2457 (<https://www.nature.com/articles/s41586-021-03536-w>). STM2457 at least can be used in vivo. Does treatment of animals with this inhibitor affect Gys2 mRNA expression?

Minor points:

1) Based on the Methods section, it seems that m6A amount was quantified using the Epigentek m6A quantification kit. This kit calculates the amount of m6A in a given amount of RNA. It cannot calculate an m6A/A ratio as seems to be indicated on the axis titles of these figures. m6A/A can only be determined by LC-MS of digested nucleotides run with proper nucleotide standards. Please confirm that the analysis has been performed accurately and according to kit specifications. An axis title of "% m6A in RNA" will be appropriate for the data presented in Fig 1c, 6b etc.

Reviewer #2 (Remarks to the Author):

Key results

The authors present compelling evidence that m6A regulation in the liver of mice is important for the proper storage of glycogen. Stabilization of the mRNA (*Gys2*) that encodes glycogen synthase 2 seems to be dependent on m6A regulation and in the absence of m6A regulation overexpression of the mRNA recovers some of the glycogen storage defects they observe. In what seems to be a unconnected story, they also show that in mice and rats glycogen and glucose storage levels are different between young and adult animals. In addition, the amount of m6A increases in liver RNA as the animals age. The manuscript requires major revisions and a few additional experiments.

Validity/ Data and methodology

The experiments seems to be appropriate to draw the conclusions the authors are making. There are however additional analysis and experiments that would strengthen the manuscript and a few places where the authors make conclusions that are not supported by their data. Histology of mice and rat are outside of my expertise.

The meRIP-seq data are very weak as presented. Only because of the authors additional experiments does their main conclusion seem convincing. As presented, I do not think they demonstrate that *Gys2* methylated. Fig3a: is not legible. It would be helpful to also know additional information, such as how many of the transcripts contain this motif. Fig3b. There needs to be an explanation in the methods how this plot was generated. It seems very strange that there are 0 5'UTR peaks. Additional controls/analysis should be presented, such as CIMS analysis to assess the quality of the data and a few examples of known methylations (*Fasn*). Furthermore, additional replicates should be added for this data to be credible. Fig3g. These data need to be normalized in order to make the comparison the authors are trying to make. I am very surprised by the number of meRIP-seq targets identified. Either this is a poor quality data set or the data analysis is overly stringent. There needs to be a much better explanation of the data analysis and a discussion of this in the text. Finally there should be a justification for selecting *Gys2* out of the 27 possible genes to study and perhaps say something about the rest of the genes in the list.

Throughout the manuscript the authors refer to HET as heterogeneous. This seems to be a typo. Do they mean heterozygous? If this is not a typo it significantly changes the meaning/interpretation of the results.

Supplementary Fig. 1c. shows the ratio of m6A to A for control (WT?) vs. conditional knockout of Mettl3 mouse livers. Since the authors use the HETs as controls throughout the manuscript it is essential to show that m6A levels are higher in these animals versus the Mettl3 conditional knockout.

Figure 4f. The most convincing and critical experiment in the manuscript is weakened by mutating two bases. The authors should discuss why they mutated GA and not just the m6A site. Mutating both bases may prevent binding of YTH and have nothing to do with m6A. They should also report whether or not they sequenced their construct and present a detailed schematic of the construct. Furthermore, while the conservation of this site in closely related animals is interesting, it would be more compelling to know that the m6A site they mutate is the one they identified in their MeRIP-seq data.

There is no source data file underlying the figures.

Not all of the sequencing data are uploaded to GEO. (The data uploaded to GEO includes 12 samples (6 IP and 6 Input). The documentation says the data were generated by MeRIP-Seq. This means there are three replicates, but the manuscript says there is one MeRIP-seq data set. There are no RNA-seq data.)

The authors say in lines 185-186, that "MeRIP-qPCR confirmed that Gys2 mRNA was an m6A-regulated target". First, if Gys2 is methylated does not mean m6A is a regulatory mark, and second, a different assay should be used to validate the MeRIP-seq data. These assays (MeRIP-seq and MeRIP-qPCR) are fundamentally the same assay. A different assay such as SCARLET (PMID: 24141618) or similar should be used to validate the RIP.

The model does not seem to come from the data that authors present. They do not show that there are equal levels transcription of Gys2 mRNA in pups and adults, nor do they show that methylation increases in Gys2 mRNA specifically in adult. Furthermore, they show data that suggests that by simply overexpressing Gys2 mRNA Fig. X, you are able to overcome/circumvent the necessity of m6A/IGF2BP2 stabilization of the message.

In line 235 the authors said, "reconstitution of GYS2 activation reversed Mettl3-cKO-associated glycogen deficiency". Their data do not support this claim. They convincingly show that overexpression of GYS2 partially restores glycogen levels, but they do not see a reversal.

The authors conclude in lines 192-293, "And loss of m6A modifications of Gys2 mRNA perish[?] its expression both in RNA and protein levels". They do not have data to backup this claim. The reduction of Gys2 mRNA and protein levels could be an indirect result of removing all mRNA methylation in these cells.

Title: M6A does not govern liver glycogenesis simply through one mRNA (Gys2). Since the authors report a partial rescue of glycogen accumulation upon overexpression to Gys2 there must be other factors involved in the accumulation of glycogen.

Supplementary Fig. 1 (line 156) when it should be 1A?

Supplementary Fig. 1b is never mentioned in the text.

Line 162 states that AQP8 was "downregulated". This implies a specific control mechanism which is not supported by the data. Reduced level would be sufficient language and not misleading.

Fig. 4 It is not clear in which system the authors conducted these experiments. Kidney cells or mouse liver? Needs to be more explicit in the results and in the figure legend.

Fig. 4D legend needs to be clearer. I am assuming the authors are using an antibody against IGF2BP2, but it is not in the legend or the results section. This assay also needs a positive and negative control. For example, probe for Fasn in the pulldown.

Supplementary Fig. 2b-f. Authors should show the actual data points and not just the means.

Fig 6c-e. The transcript levels are relative to what?

In section: Reconstitution of GYS2 rescues liver glycogenesis in Mettl3-cKO mice. I do not understand what "activation" of GYS2 means. As I understand it, the authors are overexpressing GYS2 mRNA. Activation makes me think of turning on the protein (for example, through phosphorylation) not just making more.

Elucidate/justify the reasoning for using a kidney derived cell line(HEK-293T) when studying the liver.

The introduction could be improved by contextualizing the function of Gys2 in the context of glycogen storage and glucose homeostasis, in particular since Gys2 is the gene that the paper investigates most fully.

Useful discussion point would be why is there more GYS2 protein being made even though the m6A regulation is removed.

The presentation of the figures for was adequate except for Figs. 3 and 4, though there is considerable sloppiness throughout with inconsistent sizing (figures and fonts) and alignment issues. The presentation of figure 3 could be greatly improved. The fonts are unreadable for panels 3a-d and panel 3f should show the actual data points not just the means. Likewise Fig. 4 should show the data points not just the mean/median.

Analytical approach

The analytical approach is not sufficient. The authors test multiple hypotheses simultaneously in many figures so they need to correct their statistical analysis for type 1 error. The figure legends could also be improved by reporting the statistical tests used. There needs to be much more detail in the methods sections for all the analysis done, in particular for the miRIP-seq experiment.

Reviewer #3 (Remarks to the Author):

This paper provides evidence that the N6-methyl modification of glycogen synthase mRNA plays an important role in regulating the expression of glycogen synthase and therefore the level of glycogen in the liver.

1. The findings provide novel insight into the regulation of liver glycogen synthase and glycogen levels. The findings are therefore of major significance to the field.
2. The most important point of the paper, i.e. that glycogen synthase mRNA is stabilized by m6A modification, is well documented.
3. In contrast, the relationship between liver glycogen level and m6A modification, and the amount of glycogen synthase protein is not convincing. Liver glycogen levels were measured by a qualitative assay. A quantitative assay should have been used. To conclude that liver glycogen levels correlate with m6A modification, more animals of different ages should have been used. Suckling mice/rats are on a very high-fat diet (milk). Four-week-old mice are transitioning from a high-fat diet to a high carbohydrate diet. Five-week-old mice would have been more appropriate. The literature references on age versus liver glycogen levels were for different species and therefore not relevant. In a study with mice, Roesler and Khandelwal, *Diabetes* 1985; 34: 395-402, did not find an increase in liver

glycogen with age. The relationship between age and glycogen levels is not on solid ground.

4. Whether m6A correlates with glycogen synthase protein amount is a critical issue. Rather than simply showing western blot analysis with for one sample from each group, experiments should have been run that allowed statistical analysis for this important point.

5. Since glycogen synthase activity is subject to regulation by covalent modification, the paper would have been strengthened by measuring glycogen synthase activity with and without glucose-6-phosphate which completely activates the enzyme.

6. More information should have been provide about the time that blood samples were taken from the mice

7. The authors assume that blood glucose levels are reduced by m6A deficiency because liver glycogen levels are reduced. However, this was measured in the fed state. Since glycogen synthase plays an important role in lowering blood glucose levels, it seems that the absence of glycogen synthase should in crease rather than decrease blood glucose. In other words, maybe liver glycogen levels are reduced because blood glucose levels are reduced for some other reason. Since m6A clearly regulates many enzymes, it seems likely that the situation is not as simple as presented by the authors. Most likely enzymes of gluconeogenesis are affected by the status of m6A. Likewise, enzymes that utilize gluconeogenic substrates, such as pyruvate dehydrogenase, may be affected by the status of m6A. Since these factors regulate blood glucose levels, the situation may be more complicated than presented by the authors.

Dear editor,

Here is our revised manuscript NCOMMS-21-47107A-Z with a complete point-to-point response
to the reviewers' comments.

Reviewers' comments:

Reviewer #1 (Remarks to the Author):

In this manuscript, Zhang et al. find that that m6A levels in the livers of mice and rats increase
with age and that m6A regulates the expression of Gys2 mRNA through the m6A reader protein
IGF2BP2. Overall, this works needs to more clearly demonstrate the relationship between m6A,
m6A reader proteins and Gys2 expression as stated in the points below:

**Response:** We thank the reviewer for evaluating our paper carefully and giving us valuable
suggestions. We agree with the reviewer and conducted more experiments to make our conclusion
more compelling now. We hope that the reviewers will be satisfied with the revised version of our
manuscript.

Major points:

1) Liver mRNA m6A levels increase with age of animals. Why is this? A simple explanation
could be an increase in the expression of methyltransferase subunits like METTL3, METTL14 or
WTAP. It could also be due to decreased expression of demethylases like FTO or ALKBH5. The
protein expression of these key m6A machinery components should be analyzed.

**Response:** To globally analyze the expression pattern of machinery components of
methyltransferase and demethyltransferase, we analyzed the developmental dynamics of the
mouse liver transcriptome in the open database (GSE58827). We found that only *mettl3* but not
any other machinery components of methyltransferase and demethyltransferase has a significant
increase with age of mice (Figure 2a). To further confirm this finding, we performed Western blot
to test the expression pattern of *mettl3* in the protein level, As shown in the figure 2b, the protein
level of *mettl3* increases gradually from 4-week-old to 8-week-old in the mouse liver. These
results confirmed that it was reasonable to generate *Mettl3*-cKO mice in our study.

2) The Methods section lacks details about how the MeRIP-seq data was analyzed. Which
programs were used? This is key information that is required to ensure that a robust pipeline was
used. Were the peak heights normalized to input to account for changes in expression?

**Response:** In the revised method section, we showed detailed information for analysis of
MeRIP-seq and the programs we used in lines 533-549 on page 15.

And we also showed it in the following: MeRIP-Seq was performed by Cloudseq Biotech Inc.
 (Shanghai, China) according to the published procedure (Meyer et al., 2012) with slight
 modifications. Briefly, m6A RNA immunoprecipitation was performed with the GenSeq™ m6A
 RNA IP Kit (GenSeq Inc., China) by following the manufacturer’s instructions. Both the input
 sample without immunoprecipitation and the m6A IP samples were used for RNA-seq library
 generation with NEBNext® Ultra II Directional RNA Library Prep Kit (New England Biolabs, Inc.,
 USA). The library quality was evaluated with BioAnalyzer 2100 system (Agilent Technologies,
 Inc., USA). Library sequencing was performed on an illumina Hiseq instrument with 150bp
 paired-end reads. Paired-end reads were harvested from Illumina HiSeq 4000 sequencer, and were
 quality controlled by Q30. After 3’ adaptor-trimming and low-quality reads removing by cutadapt
 software (v1.9.3). First, clean reads of all libraries were aligned to the reference genome (UCSC
 MM10) by Hisat2 software (v2.0.4). Methylated sites on RNAs (peaks) were identified by MACS
 software. Differentially methylated sites were identified by diffReps. These peaks identified by
 both softwares overlapping with exons of mRNA were figured out and chosen by home-made
 scripts. GO and Pathway enrichment analysis were performed by the differentially methylated
 protein coding genes.

Finally, each meRIP-seq data had its own input RNA-seq data for normalization. Take figure 3d
 for example, four dominant meRIP peaks were marked from a to d as follows (Attached Figure 1).
 The heights of each peak and ratio between associated two peaks were listed in Attached Table 1
 below. These results demonstrated loss of peaks in HC-KO.

Attached Figure 1. m6A MeRIP-Seq (IP) and RNA-seq (Input) revealed the location of specific
 m6A peaks and expression peaks in Gys2 locus in hepatocytes of wildtype or Mettl3-cKO mice.

Attached Table 1. Peaks’ heights and their ratio in Attached Figure 1.

	Site a	Site b	Site c	Site d
IP HC-KO	0	0	106	0

IP HC-WT	76	77	176	37
Input HC-KO	25	20	8	6
Input HC-WT	105	103	105	60
HC-KO IP/Input	0	0	13.25	0
HC-WT IP/Input	0.72	0.75	1.68	0.62

3) The authors should show tracks from each replicate for the *Gys2* locus in Fig 3g. This will help
demonstrate the consistency of the phenotype between replicates. Further, the authors should
always show the input tracks as well to account for differences in expression.

**Response:** The reviewer's point is well taken. Among six samples from *Mettl3* wild-type or cKO
mice, four samples are liver tissues and two ones are primary hepatocytes. Because of limited
depth of m6A MeRIP-seq, we only detected m6A peaks in *Gys2* mRNA from hepatocyte samples
(Figure 3d). It is consistent with the fact that *Gys2* is a hepatocyte-specific gene in liver. However,
in input (RNA-seq) data of all six samples, not only hepatocytes but liver tissues in *Mettl3*-cKO
mice had lower expression than *Mettl3* wild-type ones (Figure 3d and Attached Figure 2 below).
Intriguingly, *Fasn*, another hepatocyte-specific m6A regulated gene in liver, has very similar
pattern to *Gys2* in our data (supplementary Figure 3c and Attached Figure 3 below). These results
demonstrated that *Gys2* mRNA might also be m6A methylated in hepatocytes.

Attached Figure 2. m6A MeRIP-Seq and input RNA-seq in *Gys2* locus in liver tissues of wildtype
or *Mettl3*-cKO mice.

Attached Figure 3. m6A MeRIP-Seq and input RNA-seq in Fasn locus in liver tissues of wildtype
or Mettl3-cKO mice.

New GEO dataset had been uploaded online, and the number is GSE207566
(<https://www.ncbi.nlm.nih.gov/geo/query/acc.cgi?acc=GSE207566>), secure token for reviewers:
ytereeyorfqbhyt.

4) While the IGF2BP family have been described as m6A readers, their exact role in binding m6A
is still uncertain. Despite mentioning the critical m6A readers YTHDF1-3, the authors do not
investigate whether these have any effect on Gys2 expression. This choice seems rather arbitrary.
Therefore, the authors should test whether depletion of YTHDF members influences Gys2 as well.

**Response:** According to the reviewer's suggestion, we did all mentioned experiments. To establish
a functional link between the m6A readers and Gys2, we knocked down YTHDF1/2/3 and
IGF2BP1/2/3 one by one in Hepa1-6 cells which has a relative high level of Gys2 expression.
And we found that only depletion of IGF2BP2 significantly dampens the mRNA level of Gys2
(Figure 4a, supplementary Figure 4a).

In addition, we referred to IMPC (International Mouse Phenotyping Consortium), a famous mouse
phenotype website (<https://www.mousephenotype.org/>). Data also suggested that YTHDF2 or
IGF2BP3 KO mice have no phenotype in liver and YTHDF3 or IGF2BP1 KO mice have normal
blood glucose level. So, we can exclude these four m6A readers in our study.

5) The m6A mutants in Figures 4e and 4f are in the CDS. The authors should confirm that these
are synonymous mutations and that the amino acid sequence of the proteins is not altered as this is
currently not described. This is key as changes in the amino acid sequence may influence protein
stability.

**Response:** In the former manuscript, we constructed a mutant form of Gys2 with "GGA" to
"GCT" shift as shown below (Attached Figure 4). As a result, a glycine changed to alanine. So,
we build a new synonymous mutant construct with "GGA" to "GGT" shift, however, the protein
level of exogenous Gys2 have no significant change (data not shown). So, we have to say that the

change in the amino acid sequence may influence protein stability, but we don't know the detail
mechanism.

In order to find the real m6A modification sites, we analyzed the sequences of different m6A
peaks in hepatocytes' MeRIP-seq by a SRAMP online tool (<http://www.cuilab.cn/sramp>), and we
got two candidate sites (Figure 4d). Then, we built two different mouse mutant constructs (Figure
4d). qRT-PCR (Figure 4e) and Western blot assay (Figure 4f) in Hepa1-6 cells showed that +1172,
not +2111, is the site of m6A modification. What's more, we also built one wildtype (WT) and
two mutant (Mut) Luc-Gys2 fusion constructs (Figure 4d), dual luciferase report assay confirmed
+1172 (site 1 in Mut #1) is the site of m6A modification.

**Gys2-CDS
WT**

Mouse 1380-1402 TAGACGAATC**GGACT**TTTCAACA
Rat 1380-1402 TCGACGAATT**GGACT**TTTCAACA
Cat 1380-1402 TAGACGGATC**GGACT**TTTCAACA
Dog 1380-1402 TAGACGGATT**GGACT**TTTCAACA
Rhesus monkey 1380-1402 TAGACGGATC**GGACT**TTTCAACA
Chimpanzee 1380-1402 TAGACGGATT**GGACT**TTTCAACA
Human 1380-1402 TAGACGGATT**GGACT**TTTCAACA

**Gys2-CDS
Mut**

from Human 1380-1402 TAGACGGATT**GCT**TTTCAACA

Attached Figure 4. former figure 4e.

6) The authors test whether IGF2BP2 binds Gys2 in Figure 4d by depleting the RBP and testing
interaction with Gys2. This, of course, indicates that IGF2BP2 depletion reduces the amount of
pulled down Gys2 mRNA. However, this is obviously because of reduced IGF2BP2 in the
depleted cells in the first place! A critical experiment would be if IGF2BP2 binding to Gys2 is
reduced in context of METTL3 depletion or in the cKO. This would demonstrate whether or not
IGF2BP2 interaction with Gys2 mRNA is dependent on m6A modification. Immunoblots of Input
and IP fractions should be shown for RIP experiments to accurately draw conclusions.

**Response:** According to the reviewer's suggestion, we did the RIP-qPCR assay in the Mettl3-HET
and Mettl3-cKO hepatocytes, respectively. As shown in the revised Figure 4c, Gys2 mRNA was
enriched by IGF2BP2 antibody in HET cells while this enrichment was dampened in cKO cells.
And the result of immunoblot of input and IP fractions were shown in the supplementary Figure
4d.

7) The authors demonstrate that depletion of IGF2BP2 slightly reduces the stability of Gys2
mRNA. However, is this effect dependent on m6A? The authors should perform similar
experiments with the m6A-mutant generated in figure 4e/f in the context of IGF2BP2 depletion
(assuming they are synonymous mutants).

**Response:** According to the reviewer's suggestion, we did mRNA stability analysis in Hepa 1-6
cells, and found that depletion of IGF2BP2 robustly reduces the stability of Gys2 mRNA (Figure
4b). Meanwhile, IGF2BP2-RIP-qPCR assay in Hepa 1-6 cells, based on constructs and primers
shown in Figure 4d. It is wild type, not m6A site mutant #1, Flag-Gys2 mRNA was diminished
with IGF2BP2 depletion in RIP-qPCR assay (Figure 4h, supplementary Figure 4c). These results
demonstrated that IGF2BP2 protein binds to Gys2 mRNA in an m6A dependent manner.

8) Given that the difference in Gys2 mRNA stability is slight, it is possible that affecting m6A
pathways indirectly influences Gys2 mRNA by regulating a transcription factor instead. 4sU pulse
chase experiments will determine differences in Gys2 transcription following METTL3 or
IGF2BP2 depletion and can also provide insights into differential stability.

**Response:** Firstly, we did did mRNA stability analysis in Hepa 1-6 cells, and found that depletion
of IGF2BP2 robustly reduces the stability of Gys2 mRNA (Figure 4b).

Second, for transcription factor issue, given that 4-thiouridine (4sU) has effects on rRNA
synthesis and causes a nucleolar stress response (RNA Biol. 2013 Oct;10(10):1623-30.), we
detected nascent mRNA of Gys2 in different genotype mouse livers, another direct assay to
answer this question. As the results shown in Figure 3f and supplementary Figure 3d, mature
mRNA of Gys2 was much lower in Mettl3-cKO livers than Mettl3-HET ones, however, Gys2
nascent mRNA had almost the same expression level in Mettl3-HET and cKO livers. Similar
results were also observed in STM2457 (catalytic inhibitor of METTL3) treatment assay
(supplementary Figure 3e-f). Taken together, depletion or pharmacological inhibition of METTL3
had no significant effect on transcription of Gys2 mRNA. Stability of Gys2 mRNA still should be
the main concern in this study.

9) In Figure 4f, what are the RNA levels of the FLAG-tagged WT or m6A-mutant Gys2 constructs?
This would help to confirm whether differences in RNA stability/expression are m6A-dependent.

**Response:** Except for western blot and dual luciferase report assays in Figure 4f-g, in the revised
Figure 4e, exogenous FLAG-tagged WT Gys2 construct had higher RNA level than the
m6A-mutant one (Mut #1) in Hepa 1-6 cells. This result demonstrated that the differences in RNA
stability/expression should be m6A-dependent.

10) Recently, several small molecule inhibitors of the methyltransferase complex have been
discovered, including the older DAA and the newer STM2457
(<https://www.nature.com/articles/s41586-021-03536-w>). STM2457 at least can be used in vivo.
Does treatment of animals with this inhibitor affect Gys2 mRNA expression?

**Response:** According to the reviewer's suggestion, eight-week-old male mice were sacrificed after
i.p. treatment with vehicle or 50 mg/kg STM2457 each day for 5 days (supplementary Figure 2a).

PAS staining (supplementary Figure 2c), transmission electron microscopy (supplementary Figure
2d) and glycogen content assay (supplementary Figure 2e) showed that STM2457-treated mice
had much less glycogen in liver tissues. In addition, relative fold of m6A mRNA and Gys2 mRNA
level were both diminished in hepatocytes of STM2457-treated mice (supplementary Figure 2b,
supplementary Figure 3e).

Minor points:

1) Based on the Methods section, it seems that m6A amount was quantified using the Epigentek
m6A quantification kit. This kit calculates the amount of m6A in a given amount of RNA. It
cannot calculate an m6A/A ratio as seems to be indicated on the axis titles of these figures.
m6A/A can only be determined by LC-MS of digested nucleotides run with proper nucleotide
standards. Please confirm that the analysis has been performed accurately and according to kit
specifications. An axis title of “% m6A in RNA” will be appropriate for the data presented in Fig
1c, 6b etc.

Response: As the reviewer said, we used the EpiQuik m6A RNA Methylation Quantification Kit
(Epigentek, # P-9005) to quantify relative fold of m6A mRNA. So, we changed the axis title to
“Relative fold of m6A mRNA” in figure 1d, 6c, supplementary figure 1c, 2b.

Reviewer #2 (Remarks to the Author):

Key results

The authors present compelling evidence that m6A regulation in the liver of mice is important for
the proper storage of glycogen. Stabilization of the mRNA (Gys2) that encodes glycogen synthase
2 seems to be dependent on m6A regulation and in the absence of m6A regulation overexpression
of the mRNA recovers some of the glycogen storage defects they observe. In what seems to be a
unconnected story, they also show that in mice and rats glycogen and glucose storage levels are
different between young and adult animals. In addition, the amount of m6A increases in liver
RNA as the animals age. The manuscript requires major revisions and a few additional
experiments.

Validity/ Data and methodology

The experiments seems to be appropriate to draw the conclusions the authors are making. There
are however additional analysis and experiments that would strengthen the manuscript and a few
places where the authors make conclusions that are not supported by their data. Histology of mice
and rat are outside of my expertise.

**Response:** We thank the reviewer for evaluating our paper carefully and giving us valuable
suggestions. We agree with the reviewer and conducted more experiments to make our conclusion
more compelling now. We hope that the reviewers will be satisfied with the revised version of our
manuscript.

1. The meRIP-seq data are very weak as presented. Only because of the authors additional
experiments does their main conclusion seem convincing. As presented, I do not think they
demonstrate that Gys2 methylated.

**Response:** This is an important and similar question from the first reviewer, and we have answered
it in Major Question 3 above.

2. Fig3a: is not legible. It would be helpful to also know additional information, such as how many
of the transcripts contain this motif.

**Response:** According to the reviewer's suggestion, we analyzed the meRIP-seq data in
hepatocytes (HC) and liver tissues (LI) from both wildtype and cKO mice. The most enriched
motif of m6A peaks and p values of these peaks were shown in revised Figure 3a. In addition, the
percentages of indicated peaks, from top to bottom in Figure 3a, were 20.17%, 24.64%, 36.51%,
22.01%, 29.25% and 15.99% in total target peaks, respectively.

3. Fig3b. There needs to be an explanation in the methods how this plot was generated. It seems
very strange that there are 0 5'UTR peaks. Additional controls/analysis should be presented, such
as CIMS analysis to assess the quality of the data and a few examples of known methylations
(Fasn). Furthermore, additional replicates should be added for this data to be credible.

**Response:** In Figure 3b, the peak density plot was visualized by R package Trumpet
(<https://github.com/skyhorsetomoon/Trumpet>). In addition, we analyzed the number of 5'UTR
peaks, and found that the percentages of these peaks were 4.56%, 5.89%, 11.48%, 11.51%,
11.60%, 11.54% (with an average of 9.43%). Meanwhile, we referred to three m6A associated
articles published recently (Figure 3a of Nature. 2021 Mar;591(7849):312-316. Figure 4a of
Nature. 2021 Mar;591(7849):317-321. Extended Data Figure 3q of Nature. 2019 Mar;
567(7748):414-419.). Similar to our result, there are only a few m6A peaks in 5'UTR.

Crosslinking induced mutation site (CIMS) analysis is a method to evaluate the mutations induced
by crosslinking. However, we used meRIP-seq, a crosslinking free method, to detect m6A
modification in this study. So, it is not necessary to concern about crosslinking associated
mutation here. What's more, as shown in Figure 3d and supplementary Figure 3c, m6A peaks in
Gys2 and Fasn (positive control) loci were analyzed, and the enrichment peaks in Gys2 locus and
Fasn locus also have similar height.

Finally, in the revised manuscript we used three samples' data from each genotype to perform
density assay in Figure 3b.

4. Fig3g. These data need to be normalized in order to make the comparison the authors are trying
to make. I am very surprised by the number of meRIP-seq targets identified. Either this is a poor
quality data set or the data analysis is overly stringent. There needs to be a much better
explanation of the data analysis and a discussion of this in the text. Finally there should be a
justification for selecting Gys2 out of the 27 possible genes to study and perhaps say something
about the rest of the genes in the list.

**Response:** In order to enhance the representativeness of the screening, here we used four pairs of
different datasets to analyze. The meRIP-seq dataset contained methylation-downregulated genes
in hepatocytes between Mettl3-cKO and Mettl3-WT mice (Fold change ≥ 2000 , P value $< 1e-15$).
The other three RNA-seq datasets came from downregulated genes in hepatocytes, male liver
tissues and female liver tissues of 8-week-old Mettl3-cKO mice, respectively. That is why there
were so many different genes between each two sets. However, Gys2 and other twenty-five genes
were still enriched by this stringent strategy, it suggested that they should be *bona fide* candidates.
We focused on Gys2 by a method of exclusion. Among these 26 candidate genes, only Mlxipl,
Egfr, Fasn and Gys2 were relative to glycogen in literature. Mlxipl is a deleted gene in
Williams-Beuren syndrome, however, glycogen storage defect and hypoglycemia (two main
phenotypes of Mettl3-cKO mice) are not symptoms of this syndrome. According to IMPC
(International Mouse Phenotyping Consortium), glycogen storage defect and hypoglycemia are
not phenotype of Egfr-KO mice. Fasn, a key enzyme in fatty acid synthesis, is thought to be less
associated with glycogen synthesis, although glycometabolism and lipid metabolism are
connected. Finally, we focused on Gys2 which is liver glycogen synthase and catalyzes the
rate-limiting step in the synthesis of glycogen. It has been reported that loss-of-function mutations
of Gys2 cause type 0 glycogen storage disease (GSD-0) in children, who have glycogen storage
defect and hypoglycemia as main symptoms. Taken together, Gys2 may play dominant role in
m6A mediated glycogen storage in liver.

5. Throughout the manuscript the authors refer to HET as heterogeneous. This seems to be a typo.
Do they mean heterozygous? If this is not a typo it significantly changes the
meaning/interpretation of the results.

**Response:** HET is abbreviation of heterozygous here, means mice with albumin-cre⁺ Mettl3^{w^t/f^l}
genotype.

6. supplementary Fig. 1c. shows the ratio of m6A to A for control (WT?) vs. conditional knockout
of Mettl3 mouse livers. Since the authors use the HETs as controls throughout the manuscript it is
essential to show that m6A levels are higher in these animals verses the Mettl3 conditional
knockout.

**Response:** In supplementary Figure 1c of former manuscript, "Control" means heterozygous mice
with Albumin-cre⁺ Mettl3^{w^t/f^l} genotype. In the revised manuscript, we showed data from WT, HET
and cKO mice in supplementary Figure 1c.

7. Figure 4f. The most convincing and critical experiment in the manuscript is weakened by
mutating two bases. The authors should discuss why they mutated GA and not just the m6A site.
Mutating both bases may prevent binding of YTH and have nothing to do with m6A. They should
also report whether or not they sequenced their construct and present a detailed schematic of the
construct. Furthermore, while the conservation of this site in closely related animals is interesting,
it would be more compelling to know that the m6A site they mutate is the one they identified in
their meRIP-seq data.

**Response:** This is an important and similar question from the first reviewer, and we have answered
it in Major Question 5 above.

8. There is no source data file underlying the figures.

**Response:** Source data of this manuscript are available in the attachment.

9. Not all of the sequencing data are uploaded to GEO. (The data uploaded to GEO includes 12
samples (6 IP and 6 Input). The documentation says the data were generated by MeRIP-Seq. This
means there are three replicates, but the manuscript says there is one MeRIP-seq data set. There
are no RNA-seq data.)

**Response:** New GEO dataset had been uploaded online, and the number is GSE207566
(<https://www.ncbi.nlm.nih.gov/geo/query/acc.cgi?acc=GSE207566>), secure token for reviewers:
ytereeyorfqbhyt.

10. The authors say in lines 185-186, that “MeRIP-qPCR confirmed that Gys2 mRNA was an
m6A-regulated target”. First, if Gys2 is methylated does not mean m6A is a regulatory mark, and
second, a different assay should be used to validate the MeRIP-seq data. These assays (MeRIP-seq
and MeRIP-qPCR) are fundamentally the same assay. A different assay such as SCARLET
(PMID: 24141618) or similar should be used to validate the RIP.

**Response:** Firstly, we did meRIP-qPCR by m6A antibody from a Magna MeRIP™ m6A Kit
(17-10499-1, Millipore), so the enriched mRNA should be methylated by m6A.

Second, as mentioned above, in order to find the real m6A modification sites, we analyzed the
sequences of different m6A peaks in hepatocytes’ MeRIP-seq by a SRAMP online tool
(<http://www.cuilab.cn/sramp>), and we got two candidate sites (Figure 4d). Then, we built two
different mouse mutant constructs (Figure 4d). qRT-PCR (Figure 4e) and Western blot assay
(Figure 4f) in Hepa1-6 cells showed that +1172, not +2111, is the site of m6A modification.
What’s more, we also built one wildtype (WT) and two mutant (Mut) Luc-Gys2 fusion constructs
(Figure 4d), dual luciferase report assay confirmed +1172 (site 1 in Mut #1) is the site of m6A
modification.

Taken together, it demonstrated that Gys2 mRNA is methylated in +1172 and this modification
plays important role on Gys2 expression.

10-2. The model does not seem to come from the data that authors present. They do not show that
there are equal levels transcription of Gys2 mRNA in pups and adults, nor do they show that
methylation increases in Gys2 mRNA specifically in adult. Furthermore, they show data that
suggests that by simply overexpressing Gys2 mRNA Fig. X, you are able to overcome/circumvent
the necessity of m6A/IGF2BP2 stabilization of the message.

**Response:** Firstly, we detected the nascent and mature mRNA of Gys2 in hepatocytes of
4-week-old and 8-week-old wildtype mice by qPCR assay. The results showed that the level of
mature Gys2 mRNA was higher in 8-week-old mouse hepatocytes than 4-week-old ones (Figure
3h), however, the nascent mRNA level of Gys2 had no significant difference between ages
(supplementary Figure 3h). Secondly, we did meRIP-qPCR assays in 4-week-old and 8-week-old
mouse hepatocytes. The result demonstrated that mature Gys2 mRNA is higher enriched in
8-week-old mouse hepatocytes compared with it in 4-week-old ones (supplementary Figure 3g).
In summary, we could get a conclusion that the nascent mRNA of Gys2 could be transcribed
similarly in hepatocytes between pups and adults, but different Mettl3 expression levels (Figure
2a-b) might lead to different m6A modification levels in nascent and mature mRNA of Gys2, then
caused different stability of mature Gys2 mRNA.

11. In line 235 the authors said, “reconstitution of GYS2 activation reversed
Mettl3-cKO-associated glycogen deficiency”. There data to not support this claim. They
convincingly show that overexpression of GYS2 partially restores glycogen levels, but they do not
see a reversal.

**Response:** We have changed words to “reconstitution of GYS2 partially reversed
Mettl3-cKO-associated glycogen deficiency” in line 275 in revised manuscript.

12. The authors conclude in lines 192-293, “And loss of m6A modifications of Gys2 mRNA
perish[?] its expression both in RNA and protein levels”. They do not have data to backup this
claim. The reduction of Gys2 mRNA and protein levels could be an indirect result of removing all
mRNA methylation in these cells.

**Response:** We have changed the words to “And loss of m6A modifications perishes Gys2 mRNA
expression in a post transcription manner.” in line 229 in revised manuscript. This is a conclusion
from the data above.

13. Title: M6A does not govern liver glycogenesis simply through one mRNA (Gys2). Since the
authors report a partial rescue of glycogen accumulation upon overexpression to Gys2 there must
be other factors involved in the accumulation of glycogen.

**Response:** Our study demonstrated that METTL3 promotes the glycogenesis in liver via
stabilizing Gys2 mRNA. However, we could not completely exclude other factors that may also
facilitate METTL3 associated glycogen storage, since liver contains various effectors to regulate
glycogenesis. Life is complicated, and it usually have diverse mechanisms to regulate important
phenotypes. For example, on core mechanism of m6a regulated macrophage activation, we found
SPRED2 was target of METTL3 (Nat Commun 2021 03 02;12(1)), however, other team found
IRAKM could be another key gene in this process (Sci Adv 2021 04;7(18)). In summary, we
could get the conclusion that Gys2 plays essential role in METTL3 mediated glycogenesis in liver,
although we could not completely exclude other factors in this pathway.

14. supplementary Fig. 1 (line 156) when it should be 1A?

**Response:** We have changed words to “supplementary Figure 1a” in line 165 in revised
manuscript.

15. supplementary Fig. 1b is never mentioned in the text.

**Response:** In the revised manuscript, “supplementary Figure 1b” is shown in line 170.

16. Line 162 states that AQP8 was “downregulated”. This implies a specific control mechanism
which is not supported by the data. Reduced level would be sufficient language and not
misleading.

**Response:** We have changed words to “...was reduced in Mettl3-cKO mice (supplementary Figure
1e).” in line 176 in revised manuscript.

17. Fig. 4 It is not clear in which system the authors conducted these experiments. Kidney cells or
mouse liver? Needs to be more explicit in the results and in the figure legend.

**Response:** In the former manuscript, we tested the Gys2 expression levels in different human
hepatocellular carcinoma cell lines and HEK-293T. We found that HEK-293T cells have the
highest protein level of Gys2 in all tested cells, so we just used this cell line to do a
proof-of-concept study.

In this revised manuscript, in order to get rid of misleading, we used Hepa 1-6 to verify the
conclusion we got in HEK-293T cells. Hepa 1-6 is a mouse hepatocellular carcinoma cell line, and
has high level of Gys2.

18. Fig. 4D legend needs to be clearer. I am assuming the authors are using an antibody against
IGF2BP2, but it is not in the legend or the results section. This assay also needs a positive and
negative control. For example, probe for Fasn in the pulldown.

**Response:** Here, we supplemented figure legend in Figure 3c. In addition, we detected Fasn in
IGF2BP2 RIP-qPCR assay, however, no significant difference was found between HET and cKO
hepatocytes (supplementary Figure 4b). Perhaps, IGF2BP2 was not the m6A reader of Fasn
mRNA in this context. So, Fasn could be a negative control in this assay. For positive control,
according to literature, we fail to find the mRNA that is wrote by METTL3 and read by IGF2BP2
on m6A in hepatocytes.

19. supplementary Fig. 2b-f. Authors should show the actual data points and not just the means.

**Response:** In the revised manuscript, we showed the actual data points in each histogram,
including the figures the reviewer mentioned.

20. Fig 6c-e. The transcript levels are relative to what?

**Response:** The transcript levels were relative to β -actin. We revised the figure legends to state the
reference in qPCR assay.

21. In section: Reconstitution of GYS2 rescues liver glycogenesis in Mettl3-cKO mice. I do not
understand what "activation" of GYS2 means. As I understand it, the authors are overexpressing
GYS2 mRNA. Activation makes me think of turning on the protein (for example, through
phosphorylation) not just making more.

**Response:** We have changed the words here to "Reconstitution of GYS2 rescues liver
glycogenesis in Mettl3-cKO mice." in the revised manuscript.

22. Elucidate/justify the reasoning for using a kidney derived cell line(HEK-293T) when studying
the liver.

**Response:** As we mentioned in Question 17 above, in the revised manuscript, we used Hepa 1-6 to
test our model. Hepa 1-6 is a mouse hepatocellular carcinoma cell line with high Gys2 expression
level. In the first manuscript, we tested the Gys2 expression levels in different human
hepatocellular carcinoma cell lines and HEK-293T. Surprisingly, we found HEK-293T had
highest protein level of Gys2, so we just used this cell line to investigate the
METTL3-IGF2BP2-GYS2 axis.

23. The introduction could be improved by contextualizing the function of Gys2 in the context of
glycogen storage and glucose homeostasis, in particular since Gys2 is the gene that the paper
investigates most fully.

**Response:** We have added this section in the revised manuscript (from Line 89 to 95, on Page 3).
We also attached the words as following,

Gys2, located at 12p12.1 in human, is conserved in chimpanzee, rhesus monkey, dog, cow, mouse,
rat, chicken, and zebrafish. The protein encoded by this gene is liver glycogen synthase (GS), a
key enzyme in glycogenesis, and catalyzes the addition of α -1,4-linked glucose to the growing
glycogen chain. Mutations in this gene cause glycogen storage disease type 0 (GSD-0) in early
childhood, with hypoglycemia and liver glycogen defect as symptoms^{1,7}. However, little is
known about regulation of Gys2 expression.

24. Useful discussion point would be why is there more GYS2 protein being made even though
the m6A regulation is removed.

**Response:** I thought that in our manuscript, we only showed the obvious decreasing of GYS2
protein in liver of Mettl3-cKO mice compared with HET mice (Figure 3g). Accordingly, we did
not find the place as the reviewer mentioned.

25. The presentation of the figures for was adequate except for Figs. 3 and 4, though there is
considerable sloppiness throughout with inconsistent sizing (figures and fonts) and alignment

issues. The presentation of figure 3 could be greatly improved. The fonts are unreadable for panels
3a-d and panel 3f should show the actual data points not just the means. Likewise Fig. 4 should
show the data points not just the mean/median.

**Response:** Following the suggestion of the reviewer, we revised all the figures in this manuscript.

26. The analytical approach is not sufficient. The authors test multiple hypotheses simultaneously
in many figures so they need to correct their statistical analysis for type 1 error. The figure legends
could also be improved by reporting the statistical tests used. There needs to be much more detail
in the methods sections for all the analysis done, in particular for the miRIP-seq experiment.

**Response:** We reanalyzed all the data in correct statistics methods and showed detailed
information in figure legends. For multiple hypotheses test, we used one-way ANOVA or
two-way ANOVA in Prism (Version 6.02). In addition, we replenished the details in the methods,
including the meRIP-seq experiment (lines 539-564 in the revised manuscript).

Reviewer #3 (Remarks to the Author):

This paper provides evidence that the N6-methyl modification of glycogen synthase mRNA plays
an important role in regulating the expression of glycogen synthase and therefore the level of
glycogen in the liver.

**Response:** We thank the reviewer for evaluating our paper carefully and giving us positive
comments and valuable suggestions. We agree with the reviewer and conducted more experiments
to make our conclusion more compelling now. We hope that the reviewers will be satisfied with
the revised version of our manuscript.

1. The findings provide novel insight into the regulation of liver glycogen synthase and glycogen
levels. The findings are therefore of major significance to the field.

**Response:** We thank the reviewer for giving us positive comments.

2. The most important point of the paper, i.e. that glycogen synthase mRNA is stabilized by m6A
modification, is well documented.

**Response:** We thank the reviewer for giving us positive comments.

3-1. In contrast, the relationship between liver glycogen level and m6A modification, and the
amount of glycogen synthase protein is not convincing. Liver glycogen levels were measured by a
qualitative assay. A quantitative assay should have been used.

**Response:** Using a glycogen content assay kit (abcam ab169558), we did quantitative assay to test
liver glycogen levels in every context we mentioned in this study (Figure 1c, Figure 2c, Figure 5c,
Figure 6b, supplementary Figure 2e).

3-2. To conclude that liver glycogen levels correlate with m6A modification, more animals of
different ages should have been used.

**Response:** The relationship between relative fold of m6A mRNA and hepatic glycogen content
was analyzed among ten different age samples (five mice were 4-week-old, the other five ones

were 8-week-old). The results show that they had positive relation with $R=0.9443$, $P < 0.0001$
(Figure 1d).

3-3. Suckling mice/rats are on a very high-fat diet (milk). Four-week-old mice are transitioning
from a high-fat diet to a high carbohydrate diet. Five-week-old mice would have been more
appropriate. The literature references on age versus liver glycogen levels were for different species
and therefore not relevant. In a study with mice, Roesler and Khandelwal, *Diabetes* 1985; 34:
395-402, did not find an increase in liver glycogen with age. The relationship between age and
glycogen levels is not on solid ground.

**Response:** In our study, we are trying to explain the biological significance of different liver
glycogen storage abilities between pups and adults. Indeed, we accurately found the liver
glycogen level was very low in suckling (2-week-old) mice (Attached Figure 5). However, we did
not use these data because 2-week-old is preweaning. Furthermore, P. FERRÉ et al found that
stomach contents were almost from chow at day 25 after birth in rat (*Reprod. Nutr. Dévelop.* 26
(1986) 619-631). And 5-week-old mice are too close to adult (6-8 weeks) ones, so it should be
reasonable to choose 4-week-old mice here.

In the study you mentioned above (*Diabetes* 1985; 34: 395-402), the authors used
C57BL/KsJ-db/+ mice as control. However, we used C57BL/6N wildtype mice in our study.
There are a lot of studies (listed below) that demonstrated the differences between C57BL mouse
sub strains, maybe the different results attributed to different background and genotypes of mice.

a. Michelle M Simon, Simon Greenaway, Jacqueline K White, Helmut Fuchs, Valérie
Gailus-Durner, Sara Wells et al. A comparative phenotypic and genomic analysis of
C57BL/6J and C57BL/6N mouse strains[J]. *Genome Biology* 2013, 14:R82.

b. Hull RL, Willard JR, Struck MD, Barrow BM, Brar GS, Andrikopoulos S, Zraika S. High fat
feeding unmask variable insulin responses in male C57BL/6 mouse substrains[J]. *J*
*Endocrinol* 2017, 233(1):53-64

c. Coleman, D.L. Obese and diabetes: Two mutant genes causing diabetes-obesity syndromes in
mice. *Diabetologia* 14, 141–148 (1978). <https://doi.org/10.1007/BF00429772>

Attached Figure 5. PAS staining of rat livers in 2 weeks old (2w) and 8 weeks old (8w).

4. Whether m6A correlates with glycogen synthase protein amount is a critical issue. Rather than
simply showing western blot analysis with for one sample from each group, experiments should
have been run that allowed statistical analysis for this important point.

**Response:** Using CCLE (Cancer Cell Line Encyclopedia) data among more than one thousand cell
 lines, we analyzed the correlation between Gys2 mRNA and known key m6A machinery
 components, including Mettl3, Mettl14, Wtap, Alkbh5 and Fto. Strikingly, Mettl3, Mettl14 and
 Wtap, but not Alkbh5 or Fto, had positive correlation with Gys2 (Attached Figure 6, below).
 According to all data here, it can be concluded that the m6A level positively correlates with the
 expression of glycogen synthase 2.

Attached Figure 6. CCLE data show mRNA relations between Gys2 and Mettl3 (a), Mettl14 (b),
 Wtap (c), Alkbh5 (d) and Fto (e).

5. Since glycogen synthase activity is subject to regulation by covalent modification, the paper
 would have been strengthened by measuring glycogen synthase activity with and without
 glucose-6-phosphate which completely activates the enzyme.

**Response:** Like many other enzymes in biochemistry, glycogen synthase 2 (Gys2) is regulated on
 several levels, including transcription activation, mRNA stability, post translation modification
 and allosteric activation. In this study, we found a METTL3-IGF2BP2-GYS2 axis that controls
 glycogen storage among pups and adults. Actually, it is an adaption regulation during for a long
 time. However, allosteric activation of Gys2 protein by glucose-6-phosphate usually happens
 within minutes, even seconds. So, we mainly focus on the regulation levels happened during a
 longer time. Here, we found that Gys2 mRNA was very low without m6A modification, both in
 condition of Mettl3 knockout (Figure 3f) and Gys2 mRNA mutation (Figure 4e). Furthermore, the
 protein of Gys2 was extremely low (less than 20%) than control group (Figure 3g and Figure 4f).
 Thus, we supposed that the m6A modification might be the key regulative step of Gys2 expression,
 although we could not completely exclude transcription of nascent RNA and modification of
 protein are other important regulation steps.

6. More information should have been provide about the time that blood samples were taken from
 the mice.

**Response:** The mice and rats in this study were housed on a 12 hours light-dark cycle (zeitgeber
 time[ZT]0-ZT24). Blood samples were taken during ZT9-11.

7. The authors assume that blood glucose levels are reduced by m6A deficiency because liver
glycogen levels are reduced. However, this was measured in the fed state. Since glycogen
synthase plays an important role in lowering blood glucose levels, it seems that the absence of
glycogen synthase should increase rather than decrease blood glucose. In other words, maybe
liver glycogen levels are reduced because blood glucose levels are reduced for some other reason.
Since m6A clearly regulates many enzymes, it seems likely that the situation is not as simple as
presented by the authors. Most likely enzymes of gluconeogenesis are affected by the status of
m6A. Likewise, enzymes that utilize gluconeogenic substrates, such as pyruvate dehydrogenase,
may be affected by the status of m6A. Since these factors regulate blood glucose levels, the
situation may be more complicated than presented by the authors.

Response:

Section A. On low Gys2 level and low blood glucose

Indeed, in the fed state, glycogen synthase plays an important role in lowering blood glucose
levels. However, like human beings, mice and rats are not always eating whole day, otherwise
they do not need to store liver glycogen. Needless glucose could all transformed to lipid or other
molecules. What's more, Jose M. Irimia et al tested blood glucose levels of LGSKO (liver
glycogen synthase knock-out) and control mice. In fed, 6-hour fast and overnight fast, the blood
glucose levels in LGSKO mice were all lower than controls (Table 2 of J Biol Chem 2010 Apr
23;285(17), below). Finally, in human beings, mutation or inactivation of Gys2 (glycogen
synthase 2, liver glycogen synthase) caused Glycogen Storage Disease 0 (GSD 0) in children. Low
blood glucose and low glycogen storage in liver are two main symptoms of this disease. In
conclusion, low Gys2 level and low blood glucose level are not contradictory in our study.

TABLE 2

Blood parameters of CN and LGSKO mice under different feeding conditions

All of the results are from 4-month-old males or from 7-month-old mice for the leptin, adiponectin, and resistin hormone levels. β -Hydroxybutyrate levels were measured as total ketone bodies. The numbers in parentheses indicate the *n* values for the given groups. The results are expressed as the means \pm S.E. ND, not determined.

	Conditional			LGSKO		
	Fed	6-h fast	Overnight fast	Fed	6-h fast	Overnight fast
Glucose (mg/dl)	123.6 \pm 4.5 (8)	134.5 \pm 10.7 (8)	93.2 \pm 6.1 (8) ^{a,b}	92.0 \pm 4.5 (8) ^c	70.6 \pm 5.0 (8) ^{a,c}	74.2 \pm 5.8 (8) ^{a,c}
Lactate (mM)	2.83 \pm 0.31 (7)	3.16 \pm 0.17 (5)	2.04 \pm 0.20 (5) ^b	2.42 \pm 0.22 (12)	2.50 \pm 0.13 (5)	2.54 \pm 0.36 (5)
Ketone bodies (mg/dl)	5.77 \pm 0.48 (15)	4.53 \pm 0.76 (6)	17.9 \pm 1.6 (13) ^a	6.84 \pm 0.46 (14)	7.43 \pm 0.94 (7)	24.4 \pm 1.6 (14) ^{a,c}
Glycerol (mg/dl)	4.76 \pm 0.47 (8)	5.18 \pm 0.80 (5)	7.53 \pm 0.58 (5) ^{a,b}	5.70 \pm 0.39 (10)	7.31 \pm 0.61 (5) ^{a,c}	7.78 \pm 0.39 (5) ^a
Nonesterified fatty acids (mM)	0.86 \pm 0.08 (7)	1.02 \pm 0.09 (10)	1.86 \pm 0.06 (11) ^{a,b}	1.04 \pm 0.07 (12)	1.60 \pm 0.14 (13) ^{a,c}	1.82 \pm 0.10 (17) ^a
Triglycerides (mg/dl)	75.0 \pm 11.9 (8)	22.0 \pm 4.2 (5) ^a	94.9 \pm 17.4 (5) ^b	65.3 \pm 4.5 (10)	36.5 \pm 7.3 (5) ^a	106.3 \pm 24.3 (5) ^{a,b}
Insulin (ng/ml)	1.59 \pm 0.29 (6)	ND	0.56 \pm 0.18 (8) ^a	0.73 \pm 0.15 (7) ^c	ND	0.21 \pm 0.04 (8) ^a
Glucagon (pg/ml)	97.4 \pm 6.0 (8)	ND	77.6 \pm 10.8 (8)	102.2 \pm 8.4 (10)	ND	76.0 \pm 10.3 (8)
Leptin (ng/ml)	ND	ND	7.34 \pm 3.71 (8)	ND	ND	7.77 \pm 3.26 (8)
Adiponectin (μ g/ml)	ND	ND	18.1 \pm 2.1 (8)	ND	ND	15.0 \pm 2.1 (8)
Resistin (pg/ml)	ND	ND	677 \pm 45 (8)	ND	ND	519 \pm 32 (8)

^a p < 0.05 vs. fed conditions.

^b p < 0.05 vs. 6-h fasted conditions.

^c p < 0.05 vs. conditional mice.

Section B. On other enzyme with m6A modification in liver

As the reviewer mentioned above, m6A modification has a lot of target molecules in liver and
other tissues. So, the Mettl3-Igf2bp2-Gys2 axis must not be the only pathway to affect liver
glycogen storage in mouse. However, we conjointly analyzed a meRIP-seq dataset from
hepatocytes and other three RNA-seq datasets from hepatocytes, male liver tissue and female liver
tissue, respectively. Under this stringent strategy, 26 candidate genes emerged, including Gys2,
Mlxipl, Egfr and Fasn. These four genes had known association with glycogen in liver in literature.
Mlxipl is a deleted gene in Williams-Beuren syndrome, however, glycogen storage defect and
hypoglycemia (two main phenotypes of Mettl3-cKO mice) are not symptoms of this syndrome.
According to IMPC (International Mouse Phenotyping Consortium), a famous mouse phenotype

website (<https://www.mousephenotype.org/>), glycogen storage defect and hypoglycemia are not
phenotype of Egfr-KO mice. Fasn, a key enzyme in fatty acid synthesis, is thought to be less
associated with glycogen synthesis, although glycometabolism and lipid metabolism are
connected. Finally, we focused on Gys2 which is liver glycogen synthase and catalyzes the
rate-limiting step in the synthesis of glycogen. As we mentioned in Section A above,
loss-of-function mutation of Gys2 cause type 0 Glycogen Storage Disease (GSD) in children, who
have glycogen storage defect and hypoglycemia as main symptoms. To sum up, the
Mettl3-Igf2bp2-Gys2 axis we demonstrated here should be the key pathway affecting glycogen
storage in liver, although it may not be the only one.

REVIEWERS' COMMENTS

Reviewer #1 (Remarks to the Author):

The authors have responded to reviewer critiques appropriately.

Reviewer #3 (Remarks to the Author):

The revised manuscript has dealt with my concerns. I find the revised manuscript acceptable for publication.

Dear editor,

Here is our revised manuscript NCOMMS-21-47107B with a point-to-point response to the
reviewers' comments.

Reviewers' comments:

Reviewer #1 (Remarks to the Author):

The authors have responded to reviewer critiques appropriately.

Response: We appreciate the Reviewer's positive comments on our efforts. Many thanks for the
Reviewer's careful evaluation of our work.

Reviewer #3 (Remarks to the Author):

The revised manuscript has dealt with my concerns. I find the revised manuscript acceptable for
publication.

Response: We appreciate the Reviewer's positive comments on our efforts. Many thanks for the
Reviewer's careful evaluation of our work.